# Multivariate Conformal Prediction using Optimal Transport

**Michal Klein**                                        *michal_klein2@apple.com*
*Apple*

**Louis Bethune**                                        *l_bethune@apple.com*
*Apple*

**Eugene Ndiaye**                                        *e_ndiaye@apple.com*
*Apple*

**Marco Cuturi**                                        *cuturi@apple.com*
*Apple*

**Reviewed on OpenReview:** *https://openreview.net/forum?id=LrXAq63eT7*

## Abstract

Conformal prediction (CP) provides distribution-free uncertainty quantification by constructing prediction sets whose validity relies on ranking conformity scores. Because ranking requires an ordering, most CP methods use univariate scores; extending them to multivariate settings, where no canonical order for vectors exists, remains challenging. We build on the theory of Monge–Kantorovich quantiles and ranks to propose a geometry-aware scalarization of vector-valued scores: we transport multivariate conformity scores to the spherical uniform distribution on the unit ball via an entropic optimal transport (OT) map and use the transported radius as a scalar score. Standard split conformal calibration then applies directly, preserving finite-sample marginal coverage. The resulting method, OTCP, produces prediction regions that adapt to the empirical geometry of the score distribution, going beyond the ellipsoidal sets imposed by norm-based scalarizations. Across a benchmark of 24 multivariate regression datasets, OTCP improves efficiency and conditional-coverage metrics mainly in low output dimensions ($d \leq 4$), while we also study the computational and statistical trade-offs involved in estimating entropic OT maps.

## 1 Introduction

Conformal prediction (CP) (Gammerman et al., 1998; Vovk et al., 2005; Shafer & Vovk, 2008) is a simple and powerful framework for uncertainty quantification that provides *distribution-free marginal coverage* under exchangeability, without imposing parametric assumptions on the data distribution. Given observed data

$$D_n = \{(x_1, y_1), \ldots, (x_n, y_n)\}, \quad \text{and a new input } x_{n+1},$$

the goal is to construct a random prediction set $\mathcal{R}_\alpha(D_n, x_{n+1})$ such that the associated response $y_{n+1}$ satisfies

$$\mathbb{P}(y_{n+1} \in \mathcal{R}_\alpha(D_n, x_{n+1})) \geq 1 - \alpha,$$

where the probability is with respect to the joint distribution of $(D_n, x_{n+1}, y_{n+1})$. This is typically achieved by computing a conformity score $S(x, y, \hat{y}) \in \mathbb{R}$, for example, a prediction error of $\hat{y}$, for each observation $(x, y)$ in $D_n$, and then ranking these scores. The conformal prediction set for the new input $x_{n+1}$ consists of all candidate responses $y$ such that their score $S(x_{n+1}, y, \hat{y})$ ranks sufficiently low relative to the empirical

distribution of scores $\{S(x_i, y_i, \hat{y})\}_{i=1}^n$ to meet the target confidence level. In recent years, CP has witnessed rapid methodological and theoretical development (Barber et al., 2023; Park et al., 2024; Tibshirani et al., 2019; Guha et al., 2024), reflecting its growing applicability to challenging learning scenarios (Straitouri et al., 2023; Lu et al., 2022). Applications span a wide range of domains, including uncertainty quantification for active learning (Ho & Wechsler, 2008), anomaly detection (Laxhammar & Falkman, 2015; Bates et al., 2021), few-shot learning (Fisch et al., 2021), time series forecasting (Chernozhukov et al., 2018; Xu & Xie, 2021; Chernozhukov et al., 2021b; Lin et al., 2022; Zaffran et al., 2022), performance guarantees for learning algorithms (Holland, 2020; Cella & Ryan, 2020), and uncertainty calibration for large language models (Kumar et al., 2023; Quach et al., 2023); see also surveys in (Balasubramanian et al., 2014; Angelopoulos et al., 2024).

**From Univariate to Multivariate Scores.** A key ingredient of CP is the notion of order: the inclusion of a candidate response depends on how its score ranks relative to past observations. Consequently, classical CP methods are primarily designed for univariate scores $S(x, y, \hat{y}) \in \mathbb{R}$. This poses a challenge for problems involving multivariate responses or vector-valued scores $S(x, y, \hat{y}) \in \mathbb{R}^d$ with $d \geq 2$, where ranking is not as straightforward as in the univariate case. While conformal prediction for multivariate responses $y \in \mathbb{R}^d$ has been widely studied, the setting where the conformity score itself is vector-valued $S(x, y, \hat{y}) \in \mathbb{R}^d$ requires additional care. Existing approaches address this by scalarizing vector-valued scores through norms (Johnstone & Cox, 2021; Messoudi et al., 2022), max-aggregation (Zhou et al., 2024), manifold-based discrepancies (Kuleshov et al., 2018), or copula-based calibration (Messoudi et al., 2021; Sun & Yu, 2023; Park et al., 2024). Beyond these approaches, a key motivation for our work is *geometry-awareness*: norm-based scalarizations (Euclidean/Mahalanobis) impose ellipsoidal level sets that may be poorly aligned with skewed, multimodal, or non-elliptical score distributions. This motivates using OT-based ranks, which induce a distribution-adaptive center-outward ordering by transporting the empirical score distribution to a symmetric reference measure.

**Ordering Score Vectors using Optimal Transport.** In this line of work, and starting with the seminal reference of Chernozhukov et al. (2017) and more generally the pioneering work of Hallin et al. (2021; 2022; 2023), multiple references have explored the possibilities offered by OT theory to define a meaningful ranking or ordering in multidimensional spaces. Simply put, the analog of a rank function computed on the data can be found in the optimal Brenier map that transports the data measure to a uniform, symmetric, centered measure of reference in $\mathbb{R}^d$. As a result, a simple notion of a univariate rank for a vector $z \in \mathbb{R}^d$ can be found by evaluating the distance of the image of $z$ (according to that optimal map) to the origin. This approach ensures that the ordering respects both the geometry, i.e., the spatial arrangement of the data and its distribution: points closer to the center get lower ranks.

**Contributions** We propose to leverage recent advances in computational optimal transport (Peyré & Cuturi, 2019), using notably entropic transport map estimators (Pooladian & Niles-Weed, 2021; Cuturi et al., 2023), to handle and rank multivariate scores. More precisely:

- We propose **OTCP**, an extension of CP techniques to multivariate score functions that leverages the OT definition of higher-dimensional quantiles. This approach offers distribution-free uncertainty sets that capture the joint behavior of multivariate predictions.

- We propose a computational implementation of **OTCP** that uses the entropic map (Pooladian & Niles-Weed, 2021), computed using sample-based solutions to the regularized OT problem (Cuturi, 2013) between scores observed on actual data and reference points sampled in the unit ball. Our approach preserves coverage guarantees while being tractable. Algorithmically, **OTCP** proceeds in four steps: (i) fit a predictor on a training split, define a vector-valued conformity score, (ii) learn an OT map from score vectors to a spherical reference distribution using an independent hold-out split, (iii) scalarize each score by the transported radius, and (iv) apply standard split conformal calibration on a calibration split. This three-way data split (train / hold-out / calibration) ensures that the learned scalarization is independent of calibration data, making the finite-sample validity guarantee transparent (Section 3.3). See Figure 1 for a schematic view of steps (ii) to (iv).

- We run and compare **OTCP** against other baselines using the benchmark of regression tasks provided by (Dheur et al., 2025) showing notable improvements in lower dimensional problems.

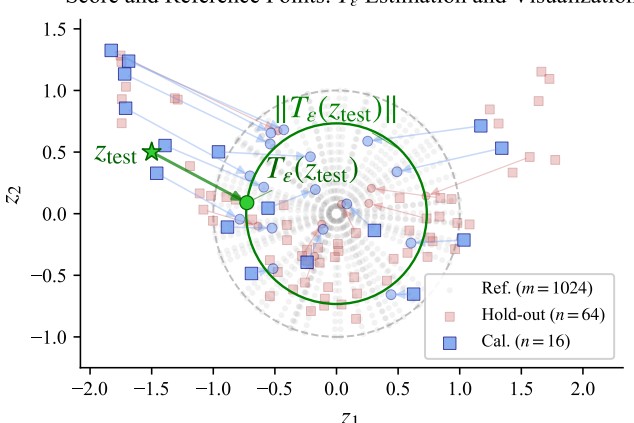 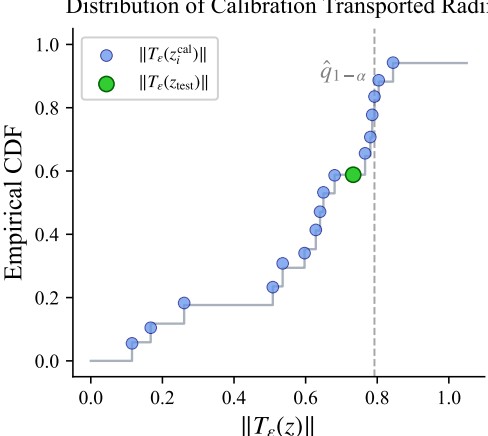

Figure 1: Schematic of the OTCP pipeline in 2D. **Left:** Score vectors from a hold-out set (red squares) are used to learn an entropic transport map $T_\varepsilon$ to a spherical reference measure (gray dots inside the unit ball). Calibration scores (blue squares) are mapped to the ball (circles) using $T_\varepsilon$ to recover conformalized vectors. Their radii $\|T_\varepsilon(z)\|$ serve as scalar conformity scores. To assess the conformality of a test score (green star), that point is sent to the ball using $\|T_\varepsilon(z)\|$. **Right:** The empirical CDF of the transported calibration radii determines the conformal threshold $\hat{q}_{1-\alpha}$; the test point is included in the prediction set if its transported radius falls below this threshold.

We acknowledge the concurrent proposal of Thurin et al. (2025); their method adopts a similar approach to ours, with, however, a few important practical differences, discussed in more detail in Section 3.1.

## 2 Background

### 2.1 Univariate Conformal Prediction

Conformal prediction is a flexible, model-agnostic framework that provides valid uncertainty quantification for machine learning models. It enables the construction of prediction sets (or intervals) that guarantee a desired coverage level such as 90% under the assumption that the data points are exchangeable which holds, for example, when the data are independently and identically distributed, or i.i.d.

To apply conformal prediction, begin by splitting your dataset into two similar but independent parts: a training set $D_{tr}$ used to fit a predictive model $\hat{y}$, and a calibration set $D_n$, used to calibrate uncertainty estimates. Once the model is trained on the training data, compute nonconformity scores $s_i = s(x_i, y_i) \in \mathbb{R}$ on the calibration set $D_n$, where $s(x, y)$ measures how atypical a label $y$ is for a new input $x_{n+1}$, given the model's prediction. From the calibration scores, determine the threshold $q_{1-\alpha}$ as the $(1-\alpha)(n+1)$-th empirical quantile of the scores $\{s_i\}_{i=1}^n$, where $n$ is the number of calibration examples. At test time, given a new input $x_{n+1}$, the conformal prediction set $\mathcal{R}(x_{n+1})$ is defined as the set of labels $y$ whose score does not exceed the quantile i.e.

$$\mathcal{R}(x_{n+1}) = \{y \in \mathcal{Y} : s(x_{n+1}, y) \le q_{1-\alpha}\} \tag{1}$$

We now state the main coverage property of conformal prediction (Vovk et al., 2005).

**Proposition 2.1.** *Let* $(X_1, Y_1), \ldots, (X_n, Y_n), (X_{n+1}, Y_{n+1})$ *be exchangeable random variables taking values in* $\mathcal{X} \times \mathcal{Y}$. *Let* $\mathcal{R}(x_{n+1})$ *be the prediction set constructed as above using a scalar valued nonconformity scores computed on a calibration set of size* $n$. *Then:*

$$\mathbb{P}(Y_{n+1} \in \mathcal{R}(X_{n+1})) \ge 1 - \alpha, \quad \forall \alpha \in (0, 1).$$

That is, the prediction set contains the true label with probability at least $1 - \alpha$, regardless of the underlying distribution, provided that the data are exchangeable. More details and proofs of this proposition can be found in the recent comprehensive review (Angelopoulos et al., 2024).

## 2.2 Multivariate Conformal Prediction

While many conformal methods exist for univariate prediction, we focus here on those applicable to *multivariate* outputs. As recalled by Dheur et al. (2025), several alternative conformal prediction approaches have been proposed to tackle multivariate prediction problems. Some of these methods can directly operate using a simple predictor (e.g., a conditional mean) of the response $y$, while some may require stronger assumptions, such as requiring an estimator of the joint probability density function between $x$ and $y$, or access to a generative model that mimics the conditional distribution of $y$ given $x$ (Izbicki et al., 2022; Wang et al., 2022). For simplicity, we restrict our attention to approaches that make no such assumption, reflecting our modeling choices for **OTCP** .

### 2.2.1 Scalarizing multivariate scores.

When the response is multivariate, a common way to apply conformal prediction is to proceed in two steps: first construct a vector-valued discrepancy $S(x,y) \in \mathbb{R}^d$, and then scalarize it through a map $\phi : \mathbb{R}^d \to \mathbb{R}$ to obtain a real-valued conformity score

$$s(x, y) = \phi(S(x, y)).$$

Standard conformal calibration is then applied to this scalar score. Existing multivariate conformal methods largely differ in either the choice of the base score $S$ (for instance residual- or quantile-based) or in the scalarization rule $\phi$. This perspective makes it possible to present M-CP, Merge-CP, and OTCP within a common framework, differing only in how the multivariate score is scalarized.

**M-CP** . We consider the template approach of (Zhou et al., 2024), which applies classical CP by aggregating a score computed on each of the $d$ outputs of the multivariate response. Given a conformity score $s_i$ for the $i$-th dimension, (Zhou et al., 2024) define the following aggregation rule:

$$s_{\text{M-CP}}(x, y) = \max_{i \in [d]} s_i(x, y_i). \tag{2}$$

Following (Dheur et al., 2025), we use *conformalized quantile regression* (CQR) (Romano et al., 2019) to define the per-output scores. For each $i \in [d]$, the conformity score is:

$$s_i(x, y_i) = \max \left\{ \hat{l}_i(x) - y_i, y_i - \hat{u}_i(x) \right\},$$

where $\hat{l}_i(x)$ and $\hat{u}_i(x)$ are conditional quantiles of $Y_i \mid X = x$ at levels $\alpha_l$ and $\alpha_u$, respectively. In our experiments we use equal-tailed intervals, $\alpha_l = \frac{\alpha}{2}$, $\alpha_u = 1 - \frac{\alpha}{2}$, where $\alpha$ denotes the target miscoverage level.

**Merge-CP** . Another widely used scalarization is to aggregate the multivariate residual into a single norm-based score,

$$s(x, y) := \|\hat{y}(x) - y\|_2,$$

with the choice of norm (e.g., $\ell_1$, $\ell_2$, or $\ell_\infty$) controlling sensitivity to errors across coordinates. This reduces the vector-valued discrepancy to a scalar score and enables direct use of standard univariate conformal calibration. A common refinement is to account for cross-target dependence through an (estimated) covariance structure, leading to ellipsoidal/Mahalanobis scalarizations (Johnstone & Cox, 2021; Messoudi et al., 2022):

$$s(x, y) := \|\Sigma^{-1/2}(\hat{y}(x) - y)\|_2,$$

where $\Sigma$ is estimated from training (or locally via kNN neighborhoods (Messoudi et al., 2022)). Related geometric score engineering has also been explored beyond second-moment structure, e.g., via manifold/Jacobian-based discrepancies (Kuleshov et al., 2018). Complementary lines of work pursue multivariate calibration by coupling multiple scalar scores through copulas (Messoudi et al., 2021; Sun & Yu, 2023; Park et al., 2024).

## 2.3 Kantorovich Ranks

A naive way to define ranks in multiple dimensions might be to measure how far each point is from the origin and then rank them by that distance. This breaks down if the distribution of the data is stretched or skewed in certain directions. To correct for this, Hallin et al. (2021) developed a formal framework of center-outward distribution functions and quantiles, also called Kantorovich ranks (Chernozhukov et al., 2017), extending the familiar univariate concepts of ranks and quantiles into higher dimensions by building on optimal transport theory.

**Optimal Transport Map and Multivariate Quantile Region**   Let $\mu$ and $\nu$ be probability measures on $\Omega \subset \mathbb{R}^d$, representing the source and target distributions, respectively. One seeks a map $T : \Omega \to \Omega$ that pushes forward $\mu$ to $\nu$ while minimizing the average transportation cost:

$$T^\star \in \underset{T_\# \mu = \nu}{\arg\min} \int_\Omega \|x - T(x)\|^2 \, d\mu(x). \tag{3}$$

Brenier (1991) ensures that if the source measure $\mu$ admits a density, there exists a solution to equation 3 that is the gradient of a convex function $\phi : \Omega \to \mathbb{R}$, i.e., $T^\star = \nabla\phi$. In the one-dimensional case, the cumulative distribution function (CDF) of a distribution $\mathbb{P}$ is the unique increasing map that transports it to the uniform distribution. This monotonicity property extends to higher dimensions through the gradient of a convex function $\nabla\phi$. Thus, the optimal transport map in multiple dimensions can be seen as a natural analog of the univariate CDF: both provide a unique, monotone transformation from one distribution to another. Following (Hallin et al., 2021; Chernozhukov et al., 2017), the center-outward distribution function of a random variable $Z \sim \mathbb{P}$ is defined as the optimal transport map $T^\star = \nabla\phi$ that pushes $\mathbb{P}$ forward to the uniform distribution $\mathbb{U}$ on the unit ball $B(0,1)$: $T^\star : \Omega \to B(0,1)$ with $T_\#^\star \mathbb{P} = \mathbb{U}$. This allows one to define the multivariate rank (and associated quantile) of a point $Z$ as $\text{Rank}(Z) = \|T^\star(Z)\|$, i.e., its radial distance from the origin. The inverse map, $(T^\star)^{-1} = \nabla\phi^*$, where $\phi^*$ is the convex conjugate of $\phi$, is referred to as the center-outward quantile map. The multivariate quantile region is a generalization of univariate quantiles to higher dimensions, representing a region in the sample space that contains a specified proportion of probability mass. The OT based multivariate quantile region is then defined as:

$$\mathcal{R}_\alpha = \{z \in \mathbb{R}^d : \|T^\star(z)\| \leq 1 - \alpha\}.$$

By construction of the spherical uniform distribution, $\|T(Z)\| \sim \text{Uniform}(0,1)$, which implies the following coverage property:

$$\mathbb{P}(Z \in \mathcal{R}_\alpha) = 1 - \alpha.$$

**Why spherical ranks?**   Following (Chernozhukov et al., 2017), we use as reference the *spherical uniform* distribution on the unit ball, obtained by drawing a direction uniformly on $\mathbb{S}^{d-1}$ and an independent radius $R \sim \text{Unif}[0,1]$. This choice provides a canonical center-outward ordering in multivariate space: if $T^*\#P = U$, then the transported radius $\|T^\star(z)\|$ is uniformly distributed on $[0,1]$, so it plays the role of a scalar multivariate rank, while $T^\star(z)/\|T^\star(z)\|$ plays the role of a multivariate sign. Small radii correspond to central points and large radii to outlying points. The spherical reference is especially convenient because its reference contours are concentric spheres, so the induced quantile and depth regions for $P$ are obtained by transporting these spheres back to the data space. This yields a geometry-aware ordering: unlike fixed norm-based scalarizations, the resulting contours can adapt to skewed, anisotropic, or even non-convex score distributions. In particular, Chernozhukov et al. (2017) show that this construction coincides with familiar notions in the univariate and spherical/elliptical settings, while remaining flexible for more general distributions. More generally, Monge–Kantorovich ranks and depth can be defined with other reference distributions $F$, such as a Gaussian law or the uniform distribution on $[0,1]^d$. However, the resulting rank, depth, and quantile regions then depend on the geometry induced by $(F, D_F)$. We adopt the spherical reference because it yields a simple one-dimensional ordering through the transported radius, is rotation-invariant, and provides a natural benchmark notion of center-outward depth. Such choice was originally recommended in (Hallin et al., 2021).

### 2.4   Entropic Map.

In practice, we do not compute the exact Brenier map $T^\star$. Instead, given score samples $z_1, \ldots, z_n$ and reference samples $u_1, \ldots, u_m$, we estimate an entropically regularized transport map $T_\varepsilon$ that is smooth, numerically stable, and valid out of sample.   A convenient estimator of the Brenier map $T^\star$ from samples $(z_1, \ldots, z_n)$ and $(u_1, \ldots, u_m)$ is the entropic map (Pooladian & Niles-Weed, 2021): Let $\varepsilon > 0$ and write $K_{ij} = [\exp(-\|z_i - u_j\|^2/\varepsilon)]_{ij}$, $i \in [n]$, $j \in [m]$, the kernel matrix. Define,

$$\mathbf{f}^\star, \mathbf{g}^\star = \underset{\mathbf{f} \in \mathbb{R}^n, \mathbf{g} \in \mathbb{R}^m}{\arg\max} \langle \mathbf{f}, \tfrac{\mathbf{1}_n}{n} \rangle + \langle \mathbf{g}, \tfrac{\mathbf{1}_m}{m} \rangle - \varepsilon \langle e^{\frac{\mathbf{f}}{\varepsilon}}, K e^{\frac{\mathbf{g}}{\varepsilon}} \rangle. \tag{4}$$

Equation (4) is an unconstrained concave optimization problem known as the regularized OT problem in dual form (Peyré & Cuturi, 2019, Prop. 4.4) and can be solved numerically with the Sinkhorn algorithm (Cuturi, 2013). Equipped with these optimal vectors, one can define the maps, valid out of sample:

$$f_\varepsilon(z) = \min_\varepsilon\left([\|z - u_j\|^2 - \mathbf{g}_j^\star]_j\right), \quad g_\varepsilon(u) = \min_\varepsilon\left([\|z_i - u\|^2 - \mathbf{f}_i^\star]_i\right), \tag{5}$$

where for a vector $\mathbf{u}$ of arbitrary size $s$ we define the log-sum-exp operator as $\min_\varepsilon(\mathbf{u}) := -\varepsilon \log(\frac{1}{s}\mathbf{1}_s^T e^{-\mathbf{u}/\varepsilon})$. Using the Brenier (1991) theorem, linking potential values to optimal map estimation, one obtains an estimator for $T^\star$ and its inverse given weights $p_j(z) := \frac{\exp\left(-(\|z-u_j\|^2-\mathbf{g}_j^\star)/\varepsilon\right)}{\sum_{k=1}^m \exp\left(-(\|z-u_k\|^2-\mathbf{g}_k^\star)/\varepsilon\right)}$,

$$T_\varepsilon(z) := z - \nabla f_\varepsilon(z) = \sum_{j=1}^m p_j(z)u_j \quad T_\varepsilon^{\mathrm{inv}}(u) := \sum_{i=1}^n q_i(u)z_i \tag{6}$$

where the weights $q_i(u)$ arise for a vector $u$ from the Gibbs distribution of $[\|z_i - u\|^2 - \mathbf{f}_i^\star]_i$.

Effectively, $T_\varepsilon(z)$ is a weighted average of the reference points $u_j$, where the weights depend smoothly on the distance between $z$ and the reference cloud. This is the map that we later use to turn a vector-valued score into a scalar radius. Note that for very small $\varepsilon$, the Sinkhorn iterations underlying entropic OT can become numerically stiff due to the sharply peaked kernel $\exp(-\|z_i - u_j\|^2/\varepsilon)$. In our use case, where $\varepsilon$ is not brought down to excessively low values, standard approaches to stabilize or speed up these computations available in the OTT-JAX toolbox (Cuturi et al., 2022) were sufficient, including Anderson acceleration (Scieur et al., 2016). Note that while there exists other solvers operating in the primal space of coupling matrices (e.g., Greenkhorn (Altschuler et al., 2017)) those cannot be used in the context of **OTCP**, because of the need to recover dual variables to form the entropic map estimator in Equation (6).

## 3 Kantorovich Conformal Prediction

### 3.1 Optimal Transport Scalarization

We consider settings where the conformity score is vector-valued, $S(x, y) \in \mathbb{R}^d$ with $d \geq 2$. In this case, the ordering used in the scalar setting no longer applies directly, and the condition $S(x_{n+1}, y) \leq q_{1-\alpha}$ in Equation (1) is no longer well defined. This motivates us to introduce *optimal transport merging*, a procedure that maps vector-valued scores $S(x, y) \in \mathbb{R}^d$ to scalar scores, allowing direct application of standard one-dimensional conformal calibration. Specifically, we define the population-level scalarization

$$S_{\mathrm{OTCP}}^\star(x, y) = \left\|T^\star\big(S(x, y)\big)\right\|,$$

where $T^\star$ is the optimal transport map pushing the distribution of scores onto the spherical reference distribution on the unit ball (cf. Equation (3)). This induces a scalar ordering through the transported radius, allowing us to return to the standard one-dimensional conformal setting. In words, OTCP first computes a multivariate score, then transports it to a spherical reference space, then keeps only the transported radius, and finally calibrates these radii by standard split conformal prediction.

Merge-CP aggregates multivariate residuals through a fixed norm (Euclidean) or a global quadratic form (Mahalanobis), which induces ellipsoidal acceptance regions in score space. In contrast, OTCP applies a distribution-adaptive transformation of the *entire* score distribution toward a spherical reference; the preimage of spherical shells under the (approximate) transport map can therefore yield score-space acceptance regions that are non-elliptical, and potentially non-convex, reflecting the empirical geometry of the score cloud (Chernozhukov et al., 2017). This suggests that OTCP is especially useful when the multivariate score distribution departs substantially from a globally ellipsoidal structure.

**Split-conformal pipeline with a learned scalarization.** Given $D_n = \{(X_i, Y_i)\}_{i=1}^n$, we split it into three disjoint sets $D_{\mathrm{train}}, D_{\mathrm{hold}}, D_{\mathrm{cal}}$. We first fit a predictor $\hat{y}(\cdot)$ on $D_{\mathrm{train}}$ and define a vector-valued score $S(x, y) := S(x, y; \hat{y}) \in \mathbb{R}^d$. We then use the hold-out scores $z_i := S(X_i, Y_i)$ for $(X_i, Y_i) \in D_{\mathrm{hold}}$ to learn the

entropic transport map $T_\varepsilon$ (Section 2.4), viewed as an approximation of the population map $T^\star$. Finally, we scalarize any candidate pair $(x, y)$ via

$$S_{\mathrm{OTCP}}(x, y) := \big\| T_\varepsilon(S(x, y)) \big\| \in \mathbb{R},$$

and calibrate this scalar score on $D_{\mathrm{cal}}$. Equivalently, OTCP can be summarized in three steps:

1. fit the predictor on $D_{\mathrm{train}}$ and define the vector-valued score $S(x, y)$;

2. learn the scalarization map $T_\varepsilon$ on $D_{\mathrm{hold}}$ and form the scalar score $S_{\mathrm{OTCP}}(x, y) = \| T_\varepsilon(S(x, y)) \|$;

3. calibrate this scalar score on $D_{\mathrm{cal}}$ using the usual split-conformal quantile.

Concretely, for $(X_i, Y_i) \in D_{\mathrm{cal}}$, we compute

$$s_i := S_{\mathrm{OTCP}}(X_i, Y_i),$$

and let $\hat{q}_{1-\alpha}$ denote the usual split-conformal empirical $(1 - \alpha)$-quantile of the calibration scores $\{s_i\}$, with the standard finite-sample correction as in Equation (1). The resulting prediction set for a new test point $x_{\mathrm{test}}$ is

$$\mathcal{R}_\alpha(x_{\mathrm{test}}) = \big\{ y \in \mathcal{Y} : S_{\mathrm{OTCP}}(x_{\mathrm{test}}, y) \leq \hat{q}_{1-\alpha} \big\}. \tag{7}$$

The additional hold-out split ensures that the learned scalarization map $T_\varepsilon$ is independent of the calibration data. Conditional on $D_{\mathrm{hold}}$, the map $T_\varepsilon$ is fixed, so $S_{\mathrm{OTCP}}(x, y)$ is simply a scalar conformity score. Therefore, the usual split-conformal finite-sample marginal validity guarantee applies directly to the prediction set in Equation (7). The price of this clean validity argument is that OTCP uses an additional hold-out split to learn the scalarization, introducing a trade-off in the allocation of data between map estimation ($D_{\mathrm{hold}}$) and threshold calibration ($D_{\mathrm{cal}}$). Thus, OT contributes the geometry-aware ordering, while conformal calibration contributes the finite-sample coverage guarantee.

This also clarifies the relation to **Merge-CP**: both methods start from a multivariate score and reduce it to a scalar, but **Merge-CP** uses a fixed norm-based scalarization, whereas OTCP learns this scalarization from the empirical score geometry. In this sense, OTCP can be viewed as extending fixed norm-based scalarizations such as **Merge-CP**. More specifically, under additional assumptions such as Gaussian-to-Gaussian or uniform-to-uniform transport, the transport map is affine (Gelbrich, 1990; Muzellec & Cuturi, 2018), so the resulting scalarization reduces to a transformed norm-based rule. The next section extends the same scalarization idea locally by learning feature-dependent transport maps, in order to better adapt the score transformation across regions of the input space.

**Comparison to concurrent work by Thurin et al. (2025).** In work developed independently and appearing on arXiv at nearly the same time, Thurin et al. (2025) proposed to leverage OT in CP with a similar approach, deriving a similar CP set and analyzing a variant with asymptotic conditional coverage under additional regularity assumptions. However, our methods differ in several key aspects. The main difference is where the OT step enters the pipeline. Thurin et al. calibrate Monge–Kantorovich ranks obtained from a discrete OT/assignment between calibration scores and reference points, together with an explicit treatment of ties. In contrast, we learn an entropic map $T_\varepsilon$ on an independent hold-out set and then apply standard split-conformal calibration to the scalar scores $\| T_\varepsilon(S(x, y)) \|$. Because the scalarization is learned on an independent split, finite-sample validity does not rely on $T_\varepsilon$ being an exact OT map. On the computational side, our implementation leverages general entropic maps (Section 3.3) without compromising finite-sample coverage guarantees. This yields a procedure that scales as $O(nm)$, decouples the number of calibration points $n$ from the target grid size $m$, and allows the map to be pre-trained once and reused at test time. In contrast, their approach requires solving a linear assignment problem, using for instance the Hungarian algorithm, which has cubic complexity $O(n^3)$ in the number of target points, and which also requires having a target set on the sphere that is of the same size as the number of input points. With our notations in Section 3.3, they require $n = m$, whereas we set $m$ to anywhere between $2^{12}$ and $2^{15}$, independently of $n$, providing smoother approximations in high dimension. Beyond computation, we also introduce localized and cluster-based variants of entropic maps to improve conditional coverage efficiently, offering a different route to adaptivity than their $k$NN-based OT-CP+. Finally, we provide extensive empirical benchmarks

and sensitivity analyses (grid size $m$, regularization $\varepsilon$) across dozens of multivariate regression tasks, giving practical guidance for using OT-based CP in real-world settings.

## 3.2 Localized Kantorovich Conformal Prediction

We generalize the OTCP score by introducing a unified framework for localizing the transport map using kernel-weighted distributions. These variants aim to better capture local structure in the residuals and improve approximation to object-conditional coverage by adapting the conformal score to the geometry of the input space. We provide a finite-sample guarantee for the hard-partition variant; kernel-weighted (soft) localization is used primarily as a practical smoothing heuristic (see Remark 3.2).

**From object-conditional to localized validity.** Formally, object-conditional validity asks (one version) that for every data-generating distribution $P$,

$$\forall P, \text{ for } P_X\text{-a.e. } x: \quad \mathbb{P}_P(Y_{n+1} \in \Gamma(D_n, x) \mid X_{n+1} = x) \geq 1 - \alpha.$$

As shown in Lei & Wasserman (2012); Vovk (2012), this distribution-free requirement is generally impossible to satisfy non-trivially on non-atomic feature spaces: allowing all $P$ permits adversarial conditional laws $P(Y \mid X = x)$ unless prediction sets become trivial. To obtain informative sets, one must therefore either weaken the conditioning (e.g., condition on partitions/taxonomies), weaken the guarantee (approximate conditional coverage), or impose structural assumptions on $P(Y \mid X)$ (e.g., smoothness/tail bounds/model correctness). Our localization strategy follows the first two routes.

Consider a kernel function $H : \mathcal{X} \times \mathcal{X} \to \mathbb{R}_+$ that measures similarity between points. Following del Barrio et al. (2024), we define a kernel-weighted empirical distribution over score vectors obtained in the hold-out set $D_{hold}$ in each region $A_k$ as

$$\mathbb{P}_{A_k} = \sum_{j=1}^{n} w_j(A_k)\, \delta_{S(X_j, Y_j)}, \qquad w_j(A_k) = \frac{H(X_j, c_k)}{\sum_{\ell=1}^{n} H(X_\ell, c_k)},$$

and $c_k$ denotes a representative point of the region (e.g., its centroid). The choice of kernel $H$ controls the weighting behavior. For example, setting $H(x, c_k) = \mathbb{1}_{x \in A_k}$ recovers uniform weights over the region (i.e., $w_j(A_k) = 1/n_k$ for $X_j \in A_k$), while a Gaussian kernel $H(x, c_k) = \exp\left(-\|x - c_k\|^2/2\sigma^2\right)$ yields soft, distance-based weighting. More generally, $H$ may incorporate adaptive metrics or local density estimates. The kernel formulation thus unifies hard and soft partitioning schemes: indicator kernels recover classical OTCP with uniform weights, while smooth kernels allow finer, geometry-aware localization within each region. Note that this input-space kernel weighting does not replace the Sinkhorn Gibbs weights $p_j(\cdot)$ in Equation (6), which remain the OT weights mapping score vectors to reference points; rather, it changes the empirical source distribution of score vectors used to estimate a local transport map. We define a single map for each region

$$S_{\text{OTCP}}^{A_k}(x, y) = \left\| T_{A_k}\big(S(x, y)\big) \right\|, \quad \forall x \in A_k.$$

**Localized split procedure.** We use a three-way split $D_n = D_{\text{train}} \cup D_{\text{hold}} \cup D_{\text{cal}}$. We fit the base predictor (and any components needed to compute $S$) on $D_{\text{train}}$. For each cell $A_k$, we estimate a local transport map $T_{A_k}$ from the hold-out score vectors $\{S(X_i, Y_i) : (X_i, Y_i) \in D_{\text{hold}}, X_i \in A_k\}$ (hard partition) or from the kernel-weighted distribution $\mathbb{P}_{A_k}$ (soft partition). Then, on $D_{\text{cal}}$, we compute calibration scores $s_i^{(k)} := \|T_{A_k}(S(X_i, Y_i))\|$ for all $(X_i, Y_i) \in D_{\text{cal}}$ with $X_i \in A_k$, and set $\widehat{q}_k$ to be the usual split-conformal $(1 - \alpha)$ empirical quantile of these scores. For a test input $x \in A_k$, we return

$$\mathcal{R}_\alpha(x) = \{y \in \mathcal{Y} : \|T_{A_k}(S(x, y))\| \leq \widehat{q}_k\}.$$

For the finite-sample guarantee below, the key requirement is that each cell contains enough calibration points so that $\widehat{q}_k$ is well-defined (in practice, $n_k^{\text{cal}} := |\{i : (X_i, Y_i) \in D_{\text{cal}}, X_i \in A_k\}|$ should not be too small). Shrinking diameters are only needed to approximate object-conditional behavior asymptotically. Under regularity conditions, the map $T_{A_k}$ pushing $\mathbb{P}(\cdot \mid A_k)$ to the reference measure converges to the conditional transport map

as $A_k \to \{x\}$ and $n \to \infty$, see (del Barrio et al., 2024, Theorem 3.2 and Corollary 3.4). Similar assumptions are required in methods like Distributional Conformal Prediction Chernozhukov et al. (2021a), where conditional coverage relies on accurate estimation of $\mathbb{P}_{Y|X=x}$. Theoretically, consistency requires the number of partitions $K$ to grow with sample size, typically under the conditions $K \to \infty$ and $K/n \to 0$. This balances two competing requirements: small regions to approximate the true conditional law, and large regions to ensure enough local samples for stability. In practice, even modest choices (e.g., $K = 5$ or $10$) already improve conditional coverage.

We now state the exact finite-sample guarantee obtained in the hard-partition case: split conformal applied within each cell yields cell-conditional coverage. Approximation enters only when one interprets increasingly fine partitions as a proxy for object-conditional coverage (Lei & Wasserman, 2012; Vovk, 2012).

**Proposition 3.1** (Finite-sample cell-conditional validity). *Assume the pairs $(X_i, Y_i)_{i=1}^{n+1}$ are exchangeable, and that the maps $T_{A_k}$ are estimated using $D_{\mathrm{hold}}$ (independent of $D_{\mathrm{cal}}$). For each $k \in [K]$, let $\widehat{q}_k$ be the split-conformal $(1-\alpha)$ empirical quantile computed from the calibration scores $s_i^{(k)} = \|T_{A_k}(S(X_i, Y_i))\|$ over $(X_i, Y_i) \in D_{\mathrm{cal}}$ with $X_i \in A_k$. Define $\mathcal{R}_\alpha(x) = \{y : \|T_{A_k}(S(x, y))\| \leq \widehat{q}_k\}$ for $x \in A_k$. Then for every $k$ such that $\mathbb{P}(X_{n+1} \in A_k) > 0$,*

$$\mathbb{P}(Y_{n+1} \in \mathcal{R}_\alpha(X_{n+1}) | X_{n+1} \in A_k) \geq 1 - \alpha.$$

*Remark* 3.2 (Soft localization). Proposition 3.1 applies directly to the hard-partition case (indicator kernel). Kernel-weighted (soft) localization can be viewed as a practical smoothing heuristic. Extending distribution-free guarantees to this setting would require controlling the likelihood ratios induced by the kernel weights, as in weighted conformal prediction (Tibshirani et al., 2019; Barber et al., 2023); formalizing such guarantees for the OT-scalarized setting is left for future work.

**Computational Trade-offs** Localizing the OTCP score either through clustering or kernel-based weights introduces a trade-off between conditional accuracy and computational cost. Hard clustering (e.g., $K$-means) is particularly efficient: given $N$ residuals and a reference distribution of size $M$ (e.g., $M = 8192$), a global Sinkhorn map costs $O(NM)$. Partitioning the data into $k$ clusters of sizes $N_1, \ldots, N_k$ leads to a total cost $\sum_{i=1}^k O(N_i M) = O(NM)$, matching the global cost while producing localized maps better suited to capture input-space heterogeneity. In contrast, soft localization solves $k$ OT problems over the full dataset, each reweighted by a kernel centered at a representative point. Each problem still incurs $O(NM)$ cost, yielding an overall complexity of $O(kNM)$ scaling linearly with $k$. Although more expensive, soft methods are trivially parallelizable: both the kernel weights and Sinkhorn maps for different centers can be computed independently. A third option is pointwise localization at test time, where a new transport map is computed from the $K$ nearest training points for each test input. This highly adaptive method, similar to that proposed in del Barrio et al. (2024), incurs $O(KM)$ per test query. However, in our experiments, it brought no notable gains over hard clustering. In summary, hard clustering strikes a practical balance: it improves conditional adaptation without exceeding the global cost, and remains especially effective in high dimensions where fine-grained structure is hard to estimate globally.

## 3.3 Implementation with the Entropic Map

We assume access to residuals samples $(z_1, \ldots, z_n)$, and a discretization of the uniform grid on the sphere, $(u_1, \ldots, u_m)$, with sizes $n, m$ that will be usually different, $n \neq m$. Learning the entropic map estimator as in Section 2.4 requires running the Sinkhorn algorithm for a given regularization $\varepsilon$ on a $n \times m$ cost matrix. At test time, for each evaluation, computing the Gibbs weights requires computing the distances of a new score $z$ to the uniform grid. The complexity is therefore $O(nm)$ when training the map and conformalizing its norms, and $O(m)$ to transport a conformity score for a given $y$.

**Sampling on the sphere.** Following Hallin et al. (2021), we begin by constructing the target distribution $\mathbb{U}_{n+1}$ as a discretized version of a spherical uniform distribution. It is defined such that the total number of points $n + 1 = n_R n_S + n_o$, where $n_o$ points are at the origin: $n_S$ unit vectors $\mathbf{u}_1, \ldots, \mathbf{u}_{n_S}$ are uniform on the sphere and $n_R$ radius are regularly spaced as $\left\{\frac{1}{n_R}, \frac{2}{n_R}, \ldots, 1\right\}$. The grid discretizes the sphere into layers of concentric shells, with each shell containing $n_S$ equally spaced points along the directions determined by the unit vectors. The discrete spherical uniform distribution places equal mass over each point of the grid, with $n_o/(n+1)$ mass on the origin and $1/(n+1)$ on the remaining points. This ensures isotropic

sampling at fixed radius onto $[0, 1]$. Additionally, we borrow inspiration from the review provided by Nguyen et al. (2024) and pick their *Gaussian-based* mapping approach (Basu, 2016). This consists of mapping a low-discrepancy sequence $w_1, \ldots, w_L$ on $[0, 1]^d$ to a potentially low-discrepancy sequence $\theta_1, \ldots, \theta_L$ on $\mathbb{S}^{d-1}$ through the mapping $\theta = \Phi^{-1}(w)/\|\Phi^{-1}(w)\|_2$, where $\Phi^{-1}$ is the inverse CDF of $\mathcal{N}(0, 1)$ applied entry-wise.

### 3.4 Differences with Vector Quantile Regression (VQR)

VQR methods (Carlier et al., 2016; Rosenberg et al., 2022; Pegoraro et al., 2023) aim to estimate the full conditional quantile map via optimal transport under a mean-independence constraint, extending earlier $L_1$-based approaches (Chaudhuri, 1996). In contrast, our framework generalizes conformal prediction to vector-valued non-conformity scores without assuming access to a conditional density model. VQR formulates quantile regression as a large-scale linear program over a transport plan $\Pi \in \mathbb{R}^{T^d \times N}$ subject to $\Pi \mathbf{1}_N = \frac{1}{T^d} \mathbf{1}_{T^d}$, and $\Pi^\top \mathbf{u} = \bar{X}$, where $\mathbf{u} \in \mathbb{R}^{T^d \times d}$ is the grid of quantile targets and $\bar{X}$ the empirical mean of covariates. The number of variables scales as $T^d N$, which becomes prohibitive beyond $d = 2$; in practice, the (NL)VQR implementation hardcodes $T^d \approx 8000$. Dual relaxations do not yield explicit quantile maps, and enforcing monotonicity requires Vector Monotone Rearrangement (VMR) (Rosenberg et al., 2022) at test time. VQR also lacks built-in coverage guarantees and typically relies on scalar conformalization over post-estimated regions (Feldman et al., 2023), e.g., $S(x, y) = \max(\text{dist}(y, R(x)), \text{dist}(y, R^c(x)))$. By contrast, OTCP estimates a transport map from residuals to a continuous reference via Sinkhorn regularization with complexity $O(NM)$, avoiding gridding and scaling well with dimension. In our evaluation, we matched the number of OTCP target points to the VQR grid ($T^d \approx 8000$) and found OTCP outperformed VQR in coverage tasks. This should not be seen as a critique of VQR's modeling power, but as evidence that its generality does not necessarily yield practical benefits for calibrated prediction in higher dimensions.

## 4 Experiments

### 4.1 Setup and Metrics

We borrow the experimental setting provided by Dheur et al. (2025) and benchmark multivariate conformal methods on a total of 24 tabular datasets. Total data size $n$ in these datasets ranges from 103 to 50,000, with input dimension $p$ ranging from 1 to 348, and output dimension $d$ ranging from 2 to 16. We adopt their approach, which is to rely on a multivariate quantile function forecaster (MQF$^2$, Kan et al., 2022), a normalizing flow that is able to quantify output uncertainty conditioned on input $x$. However, in accordance with our stance mentioned in the background section, we will only assume access to the conditional mean (point-wise) estimator for **OTCP**. If one has access to a smooth joint or conditional distribution $P_{Y|X}$, one may define vector-valued scores using samples from $P_{Y|X}$ (as in PCP), or extract a conditional map $T_x \# P_{Y|X=x} = \mathbb{U}$ to a reference distribution. Our framework remains compatible with such maps: if $T$ pushes $Z$ to a reference, then any invertible function $f$ induces a new map $T_f = f \circ T \circ f^{-1}$, allowing transport on conformity scores $s(x, y)$ via composition. While promising, estimating such conditional maps can be computationally intensive and is left for future work.

We incorporated several vector quantile regression (VQR) baselines into the evaluation: `ST-DQR-CP`, `VQR`, `NL-VQR`, `VQR-CP`, and `NL-VQR-CP`. We also implemented a simple localization strategy for OTCP based on $k$-means clustering, resulting in two variants: `OTCP-CLS (5)` and `OTCP-CLS (10)`, where the number in parentheses indicates the number of clusters used. Furthermore, we report two conditional coverage metrics to assess the quality of local calibration. As is common in the field, we evaluate the methods using several metrics, including marginal coverage (MC), and mean region size (Size). The latter is using importance sampling, leveraging (when computing test time metrics only), the generative flexibility provided by the MQF$^2$ as an invertible flow. See (Dheur et al., 2025) and their code for more details.

### 4.2 Hyperparameter Choices

We apply default parameters for all three competing methods, **M-CP** and **Merge-CP**, using (or not) the Mahalanobis correction and set a target coverage $1 - \alpha = 0.8$. For **M-CP** using conformalized quantile

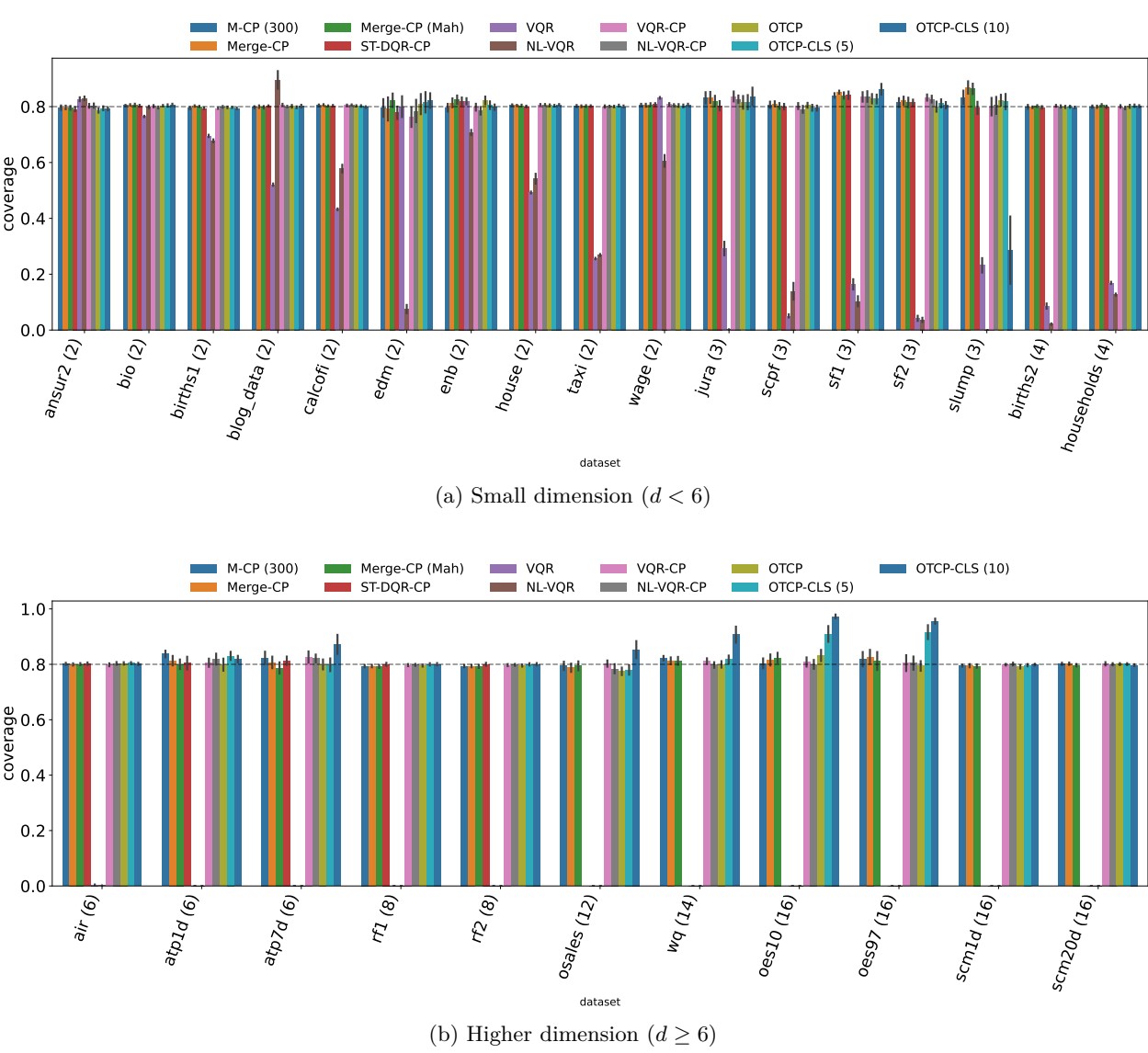

(a) Small dimension ($d < 6$)

(b) Higher dimension ($d \geq 6$)

Figure 2: Marginal coverage across datasets, split by output dimension. OTCP preserves the target coverage level comparably to standard baselines; its main differences appear in efficiency and conditional-coverage metrics rather than in marginal coverage itself.

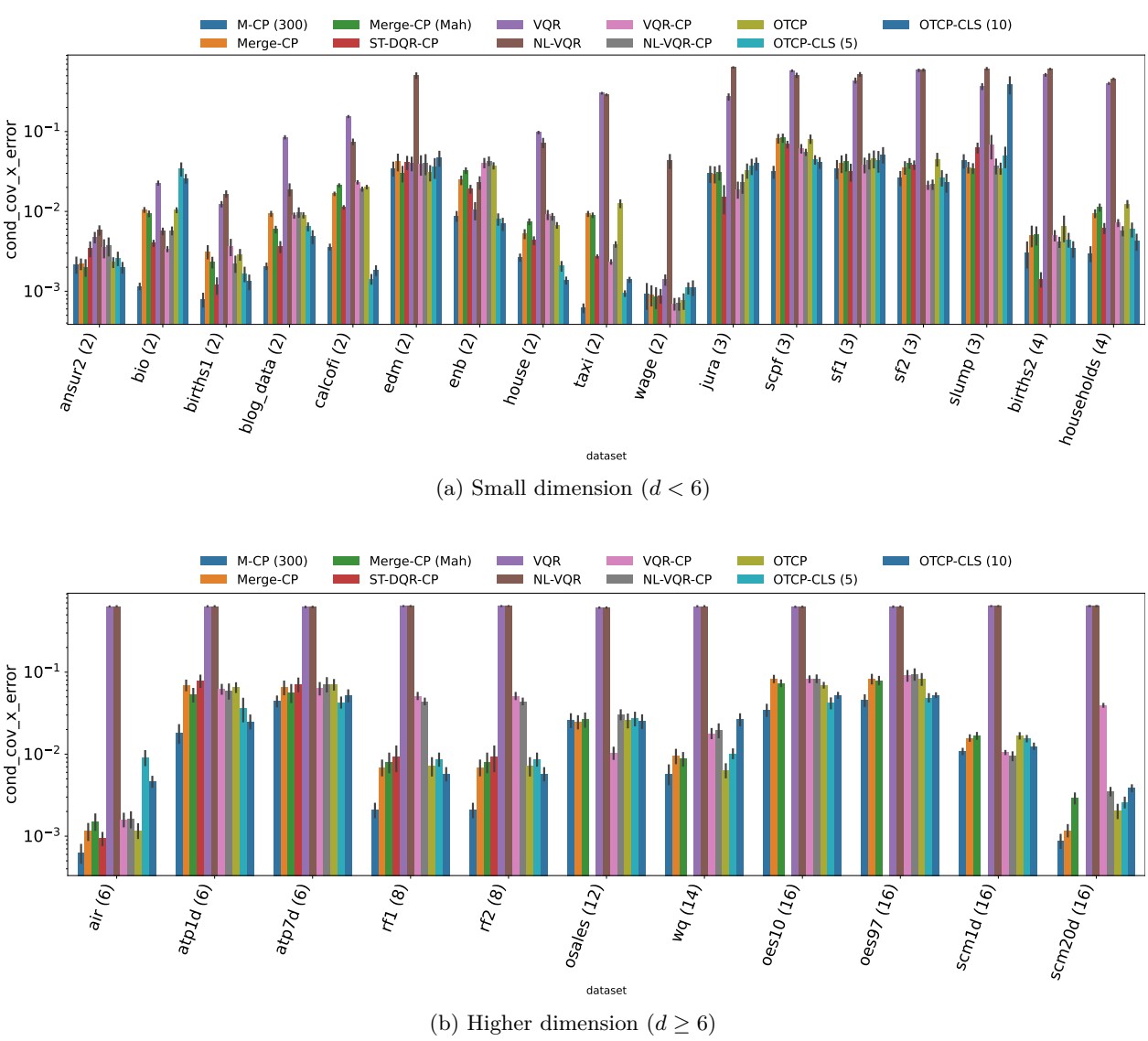

(a) Small dimension ($d < 6$)

(b) Higher dimension ($d \geq 6$)

Figure 3: Conditional coverage error (CEC-X) across datasets. Lower is better. Localized OTCP variants often reduce this error, indicating improved adaptation to heterogeneity in the feature space.

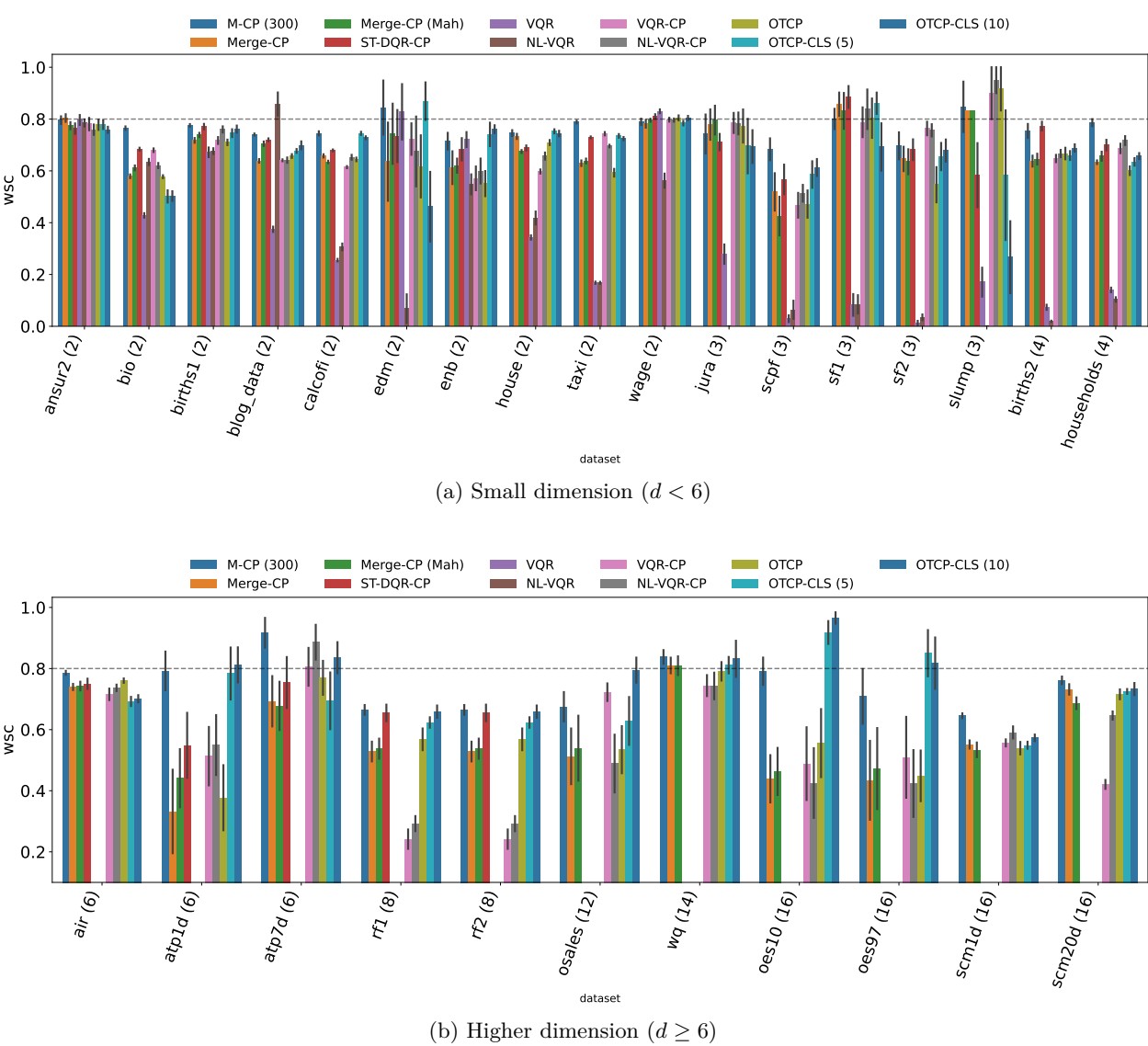

(a) Small dimension ($d < 6$)

(b) Higher dimension ($d \geq 6$)

Figure 4: Worst-case conditional coverage (WSC) across datasets, split by output dimension. Higher is better. Localized OTCP variants often improve WSC relative to global methods, suggesting that adapting the transport map across regions of the feature space can mitigate heterogeneity that is missed by a single global scalarization.

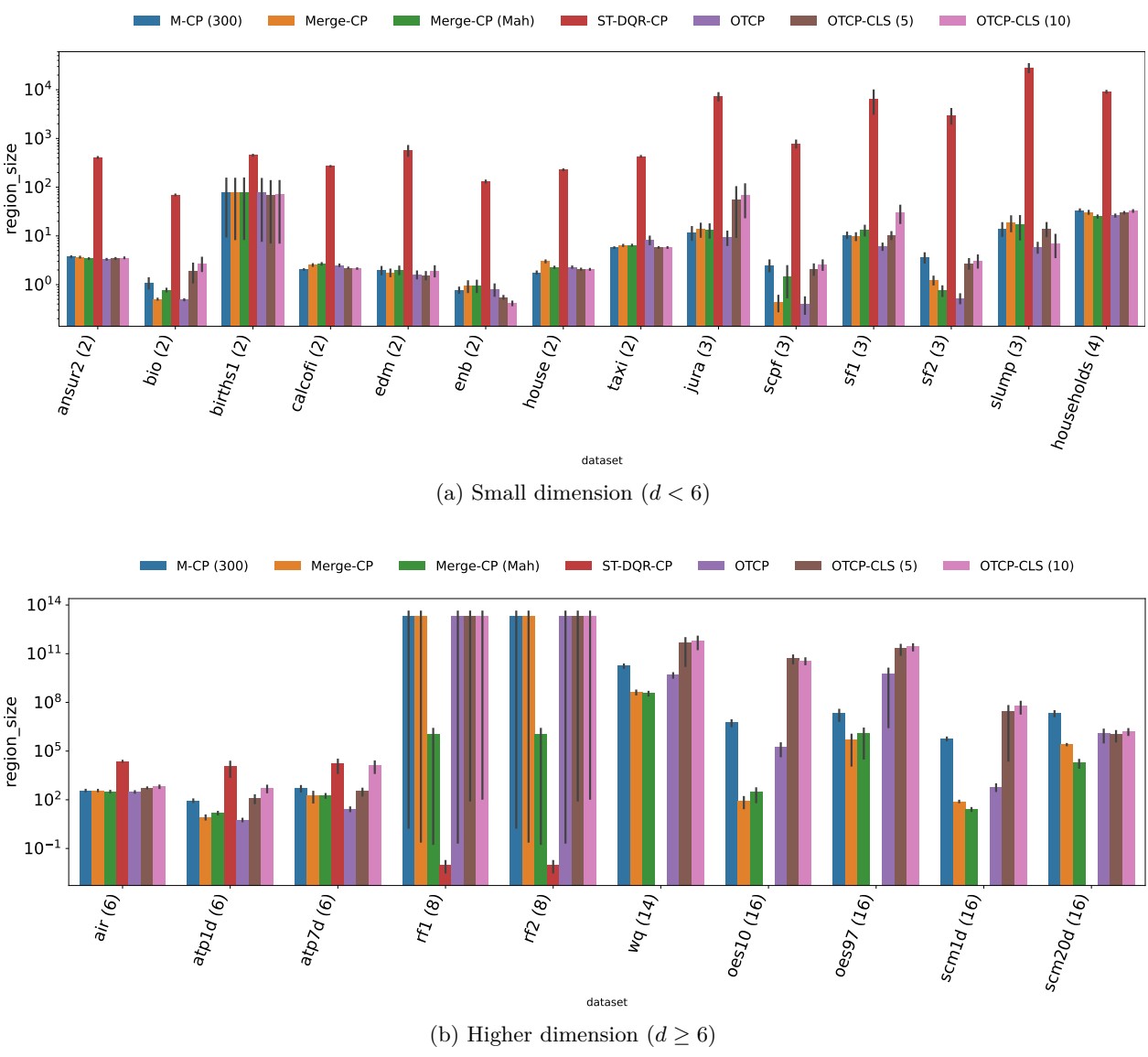

(a) Small dimension ($d < 6$)

(b) Higher dimension ($d \geq 6$)

Figure 5: Mean region size across datasets. In low dimensions, OTCP often improves efficiency relative to fixed norm-based scalarizations, whereas this advantage weakens in higher dimensions.

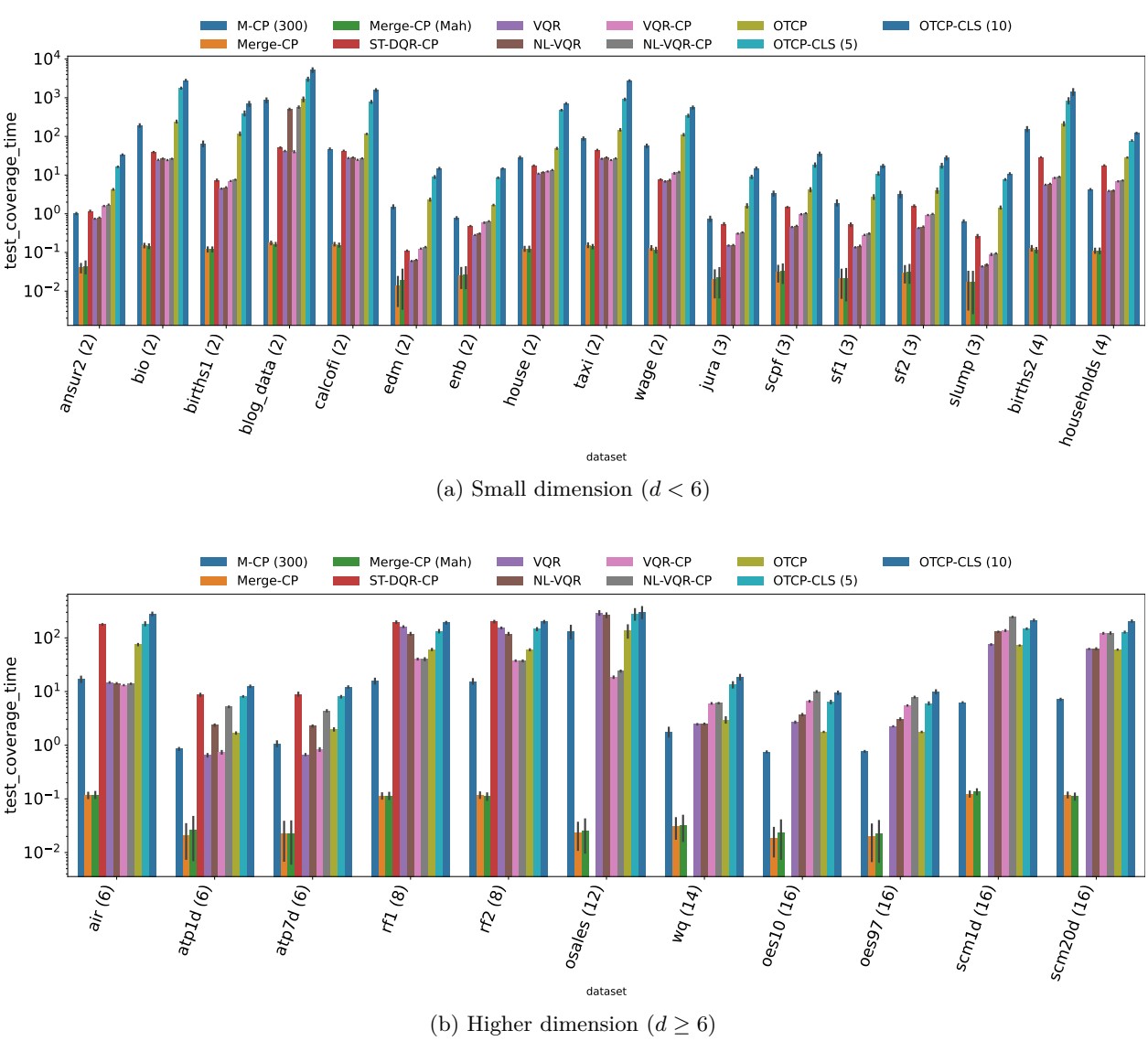

(a) Small dimension ($d < 6$)

(b) Higher dimension ($d \geq 6$)

Figure 6: Test-time computational cost across datasets, split by output dimension. OTCP incurs additional runtime relative to simple fixed-scalarization baselines because of the transport-map estimation step. This cost increases with output dimension, reflecting the added computational burden of geometry-adaptive scalarization.

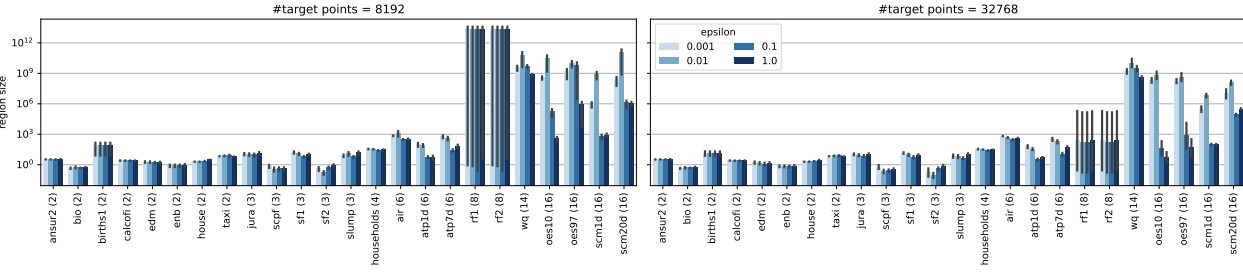

Figure 7: Sensitivity of OTCP to the number of reference points $m$ and to the regularization level $\varepsilon$. This plot details the impact of the two important hyperparameters one needs to set in **OTCP**: number of target points $m$ sampled from the uniform ball and the $\varepsilon$ regularization level. As can be seen, larger sample size $m$ improves region size (smaller the better) for roughly all datasets and regularization strengths. On the other hand, one must tune $\varepsilon$ to operate at a suitable regime: not too low, which results in the well-documented poor statistical performance of unregularized / linear program OT, nor too high, which would lead to a collapse of the entropic map to the sphere. Using OTT-JAX and its automatic normalizations, we see that $\varepsilon = 0.1$ works best overall.

regression boxes, we follow (Dheur et al., 2025) and leverage the empirical quantiles returned by $MQF^2$ to compute boxes (Zhou et al., 2024).

**OTCP**: Our implementation (coded in the OTT-JAX toolbox (Cuturi et al., 2022)) to compute these maps requires tuning two important hyperparameters: the entropic regularization $\varepsilon$ and the total number of points used to discretize the sphere $m$, not necessarily equal to the input data sample size $n$. These two parameters play complementary roles. Increasing $m$ makes the reference geometry finer and can improve the quality of the estimated map, at the cost of higher computation. Increasing $\varepsilon$ makes the map smoother and easier to estimate numerically, but if $\varepsilon$ is too large the map becomes overly blurred and loses geometric fidelity. In practice, $m$ mainly controls resolution, while $\varepsilon$ controls the bias–variance and accuracy–stability trade-off of the transport step.

On the one hand, it is known that increasing $m$ will mechanically improve the ability of $T_\varepsilon$ to recover in the limit $T^\star$ (or at least solve the semi-discrete (Peyré & Cuturi, 2019) problem of mapping $n$ data points to the sphere). However, large $m$ incurs a heavier computational price when running the Sinkhorn algorithm. On the other hand, increasing $\varepsilon$ improves *both* computational and statistical behavior, but moves the estimated map away from the ground-truth target $T^\star$ toward a more blurred map. We have experimented with these aspects and derive from our experiments that both $m$ and $\varepsilon$ should be increased to track increases in dimension. Note that the value $\varepsilon = 0.1$ reported throughout our experiments is a *relative* regularization strength: OTT-JAX (Cuturi et al., 2022) automatically rescales $\varepsilon$ by the standard deviation of the cost matrix, so the effective regularization adapts to the scale of the data. This makes our default choice portable across datasets without manual tuning of absolute scales. We also tested the debiasing procedure of Pooladian et al. (2022), i.e., centering corrections applied to regularized transport maps, but did not observe consistent improvements in our conformal prediction metrics, in agreement with the findings in (Pooladian et al., 2022).

## 4.3 Results

We present results by differentiating datasets with small dimension $d \leq 6$ from datasets with higher dimensionality $14 \leq d \leq 16$, that we expect to be more challenging to handle with OT approaches, owing to the curse of dimensionality that might degrade the quality of multivariate quantiles. Results in Figure 2a and Figure 3a indicate an improvement (smaller region for similar coverage) on 15 out of 18 datasets in lower dimensions, this edge vanishing in the higher-dimensional regime. Compared to Vector Quantile Regression approaches, we observed that VQR-based methods tend to underperform on our benchmark tasks, likely due to scalability issues and the absence of inherent coverage guarantees. In contrast, the localized versions of OTCP (`OTCP-CLS`) demonstrate improved conditional coverage, consistent with our expectations. These results confirm the benefit of incorporating even simple localization techniques into OTCP to better adapt to heterogeneous regions of the input space. Ablations provided in Figure 7 highlight the role of $\varepsilon$ and $m$, the

entropic regularization strength and the sphere size respectively. These results show that results for high $m$ tend to be better but more costly, while the tuning of the regularization strength $\varepsilon$ needs to be tuned according to dimension (Vacher & Vialard, 2022). Finally, Figure 8 provides an illustration of the non-elliptic CP regions outputted by **OTCP**, by pulling back the rescaled uniform sphere using the inverse entropic mapping.

## 5 Conclusion

We have proposed **OTCP**, a new approach that can leverage a recently proposed formulation for multivariate quantiles that uses optimal transport theory and optimal transport map estimators. We show the theoretical soundness of this approach, but, most importantly, demonstrate its applicability throughout a broad range of tasks compiled by (Dheur et al., 2025). Compared to baselines based on fixed norm-based scalarizations, OTCP is most competitive in low-dimensional settings, where it often yields smaller or better-shaped prediction regions for comparable coverage, at the cost of additional transport-map estimation time.

Our approach relies on the estimation of an optimal transport map from a finite set of score samples, a task with known statistical challenges that become acute in high dimensions (Chewi et al., 2024). As the dimension d of the score vector grows, the number of samples ($n_{\text{hold}}$) needed to faithfully represent the geometry of the score distribution increases exponentially. This curse of dimensionality directly impacts **OTCP**'s performance. Our results for datasets with $d \geq 6$ clearly illustrate this limitation. The diminishing advantage of OT-CP over simpler baselines in these settings is not a failure of the concept, but rather a reflection of the fundamental statistical cost of non-parametric map estimation. We note that the observed threshold around $d \approx 6$ is likely dependent on the sample sizes available in our benchmark (typically $10^2$–$10^4$); with substantially larger hold-out sets, OT map estimation may remain effective at higher dimensions. While we demonstrate that a pragmatic choice of hyperparameters can still yield reasonable results, this highlights a clear boundary for the current applicability of our method. Overcoming this will likely require moving beyond unstructured point clouds, for instance by exploring OT methods tailored for structured distributions or factorized assumptions, which remains a promising avenue for future research. Additionally, controlled experiments on synthetic data with known non-elliptical score distributions (e.g., multimodal or skewed) would help isolate and quantify the geometric advantage of OT-based scalarization over norm-based alternatives.

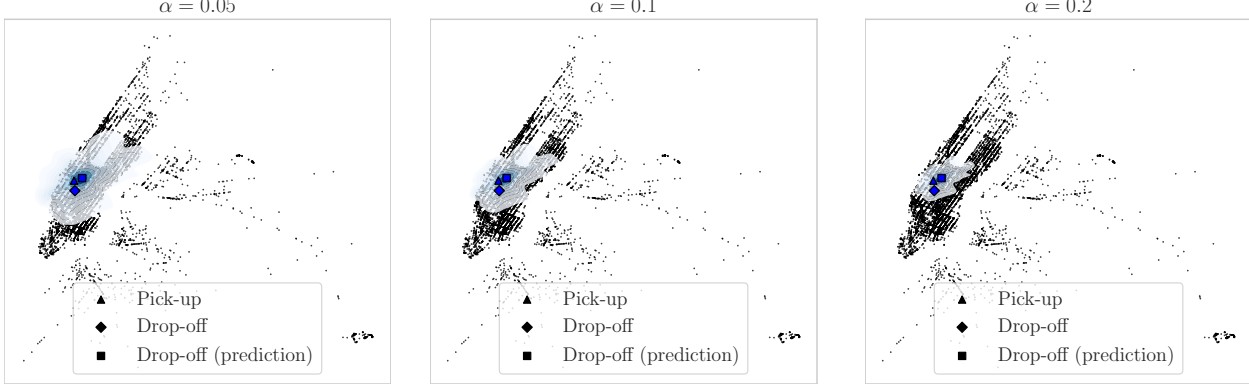

Figure 8: $K$-means localized Conformal sets, recovered by mapping back the reduced sphere on the Manhattan map, on a prediction for the `taxi` dataset. We use the inverse entropic map mentioned in Section 3.3, mapping back the gridded sphere of size $m = 2^{15}$ for each level, plotting its outer contour.

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

# A  Appendix

We conduct several additional ablation experiments to further analyze and support the main findings presented in Section 4. These experiments are designed to systematically examine the influence of key hyperparameters and experimental settings on the coverage, runtime metrics and region size across different datasets and dimensionalities. By isolating and varying specific components of our approach and baselines, we aim to provide a clearer insights into the robustness and generalizability of our methods. The results of these ablation studies are summarized in the following figures and tables.

## A.1  Benchmark Datasets

Table 1: Summary of the 24 multivariate regression datasets used in our experiments. The datasets are grouped and sorted by dimension to align with the analysis of low-dimensional ($d < 6$) versus high-dimensional ($d \geq 6$) performance.

| (a) Small Dimension Datasets ($d < 6$) | | | |
|---|---|---|---|
| **Dataset** | **Samples (n)** | **Input dim. (p)** | **Output dim. (d)** |
| ansur2 | 6068 | 98 | 2 |
| bio | 45730 | 8 | 2 |
| births1 | 1577 | 11 | 2 |
| calcofi | 50000 | 1 | 2 |
| edm | 154 | 16 | 2 |
| enb | 768 | 8 | 2 |
| house | 21613 | 14 | 2 |
| taxi | 50000 | 4 | 2 |
| jura | 359 | 15 | 3 |
| scpf | 1137 | 23 | 3 |
| sf1 | 1066 | 10 | 3 |
| sf2 | 1066 | 10 | 3 |
| slump | 103 | 7 | 3 |
| households | 7207 | 4 | 4 |

| (b) Higher Dimension Datasets ($d \geq 6$) | | | |
|---|---|---|---|
| **Dataset** | **Samples (n)** | **Input dim. (p)** | **Output dim. (d)** |
| air | 500 | 11 | 6 |
| atp1d | 337 | 411 | 6 |
| atp7d | 296 | 411 | 6 |
| rf1 | 9005 | 64 | 8 |
| rf2 | 9005 | 64 | 8 |
| wq | 1060 | 16 | 14 |
| oes10 | 403 | 298 | 16 |
| oes97 | 263 | 262 | 16 |
| scm1d | 9803 | 280 | 16 |
| scm20d | 8966 | 60 | 16 |

## A.2 Benchmark Metrics

We evaluate the conformal methods using several classical metrics that assess both coverage properties and region efficiency. We follow and refer to Dheur et al. (2025) for more details

**Region size.** Smaller regions are preferred for sharper uncertainty quantification, provided coverage guarantees are maintained. The size of a prediction region $\hat{R}(x)$ is defined as

$$|\hat{R}(x)| = \int_{\mathcal{Y}} \mathbf{1}\{y \in \hat{R}(x)\} \, dy. \tag{8}$$

Since this integral is intractable in high dimensions, we approximate it using importance sampling with the predictive density $\hat{f}(y \mid x)$:

$$|\hat{R}(x)| \approx \frac{1}{K} \sum_{k=1}^{K} \frac{\mathbf{1}\{\hat{Y}^{(k)} \in \hat{R}(x)\}}{\hat{f}(\hat{Y}^{(k)} \mid x)}, \quad \hat{Y}^{(k)} \sim \hat{f}(\cdot \mid x). \tag{9}$$

**Worst Slab Coverage (WSC).** WSC quantifies how well coverage is preserved across all directions in the input space, capturing conditional validity. For a direction $v \in \mathbb{R}^d$, the slab coverage is defined as

$$WSC_v = \inf_{a < b} \left\{ \hat{P}_{D_{\text{test}}}\left(y_i \in \hat{R}(x_i) \mid a \leq v^\top x_i \leq b\right) \; : \; \hat{P}_{D_{\text{test}}}(a \leq v^\top x_i \leq b) \geq \delta \right\}, \tag{10}$$

where $\delta \in (0, 1]$ is a minimal mass threshold. The worst-slab coverage is then

$$WSC = \min_{v_j \in S^{d-1}} WSC_{v_j}, \tag{11}$$

with $S^{d-1}$ the unit sphere, approximated by sampling random vectors $v_j$.

**Coverage Error Conditional on $X$ (CEC-X).** Partition the input space into clusters $A_1, \ldots, A_J$ (e.g., using k-means++). Then

$$CEC\text{-}X = \frac{1}{|D_{\text{test}}|} \sum_{i=1}^{|D_{\text{test}}|} \sum_{j=1}^{J} \left( \hat{P}_{D_{\text{test}}}(y^{(i)} \in \hat{R}(x^{(i)}) \mid x^{(i)} \in A_j) - (1 - \alpha) \right)^2. \tag{12}$$

*Rationale:* Measures deviations from the target coverage $1 - \alpha$ within regions of the input space.

**Coverage Error Conditional on $V$ (CEC-V).** More robust to high-dimensional inputs, since conditioning is done on predictive density rather than raw features. Define $V = \hat{f}(\hat{Y} \mid X)$ with $\hat{Y} \sim \hat{f}(\cdot \mid X)$. Construct a feature vector $v_x$ from order statistics of $\log V$, and cluster in this density space. Then compute

$$CEC\text{-}V = \frac{1}{|D_{\text{test}}|} \sum_{i=1}^{|D_{\text{test}}|} \sum_{j=1}^{J} \left( \hat{P}_{D_{\text{test}}}(y^{(i)} \in \hat{R}(x^{(i)}) \mid v_{x^{(i)}} \in A_j) - (1 - \alpha) \right)^2. \tag{13}$$

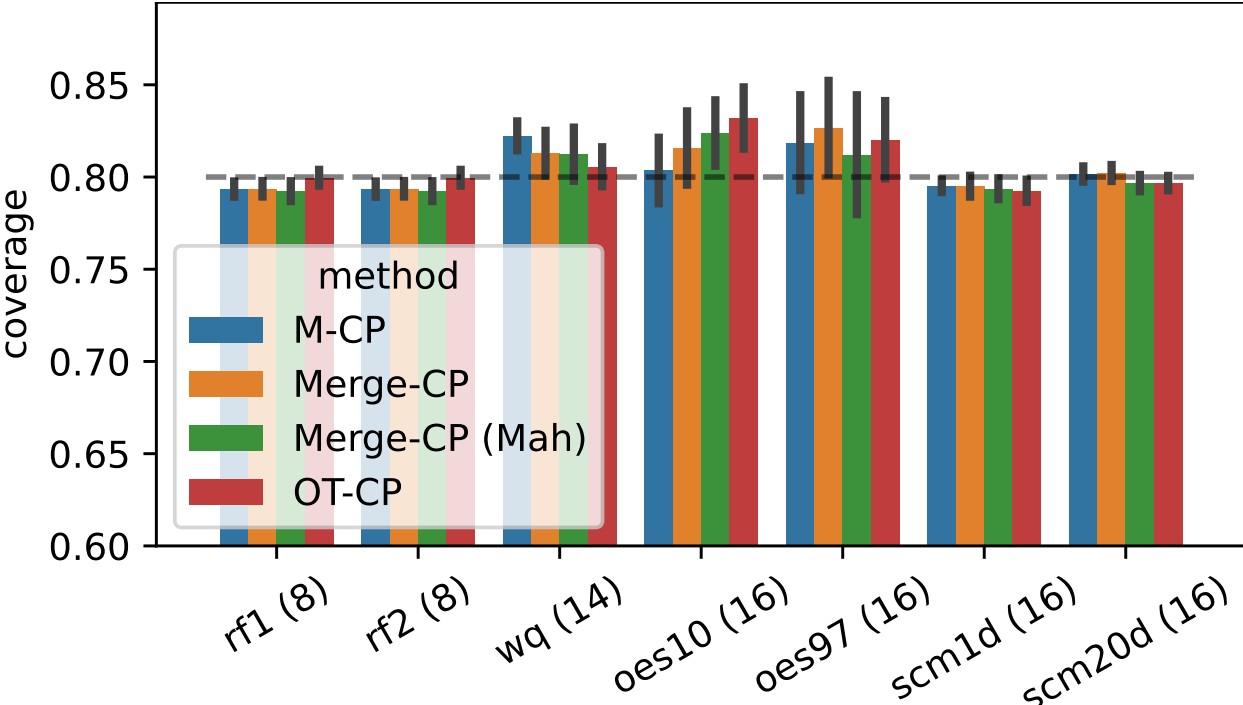

Figure 9: Coverage for higher dimensional datasets, corresponding to the setting displayed in Figure 2b.

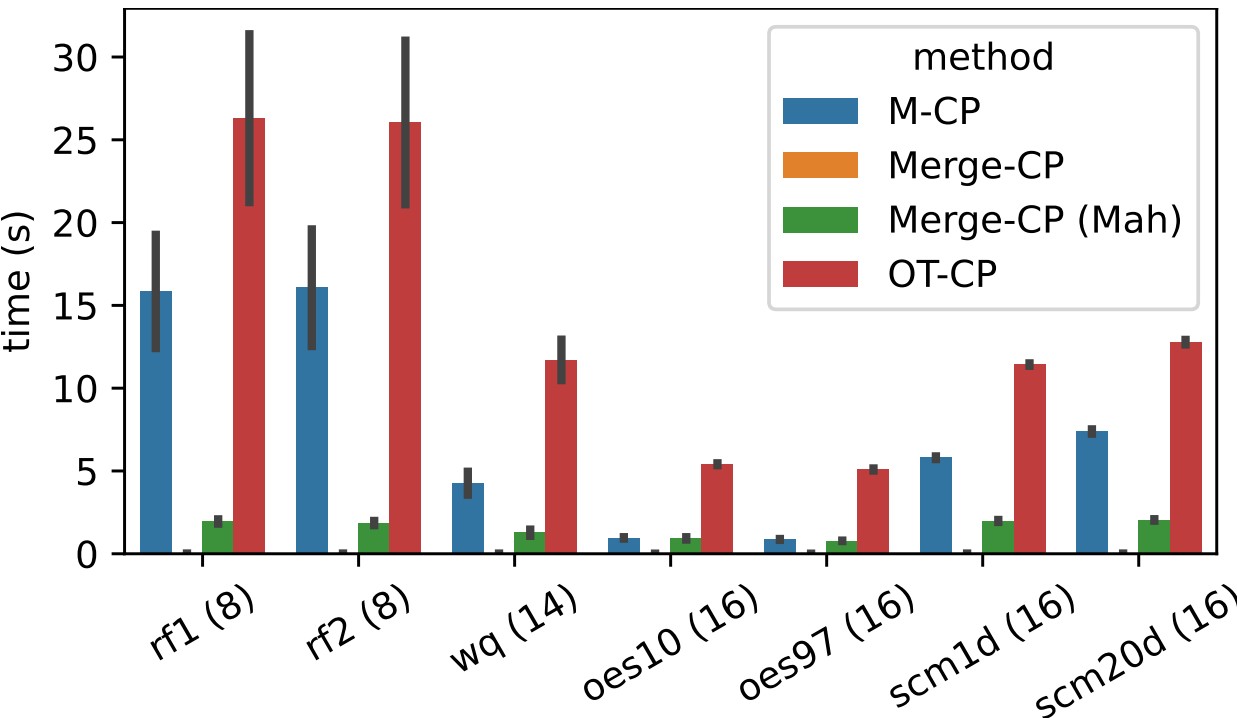

Figure 10: Runtimes for higher dimensional datasets, corresponding to the setting displayed in Figure 6b.

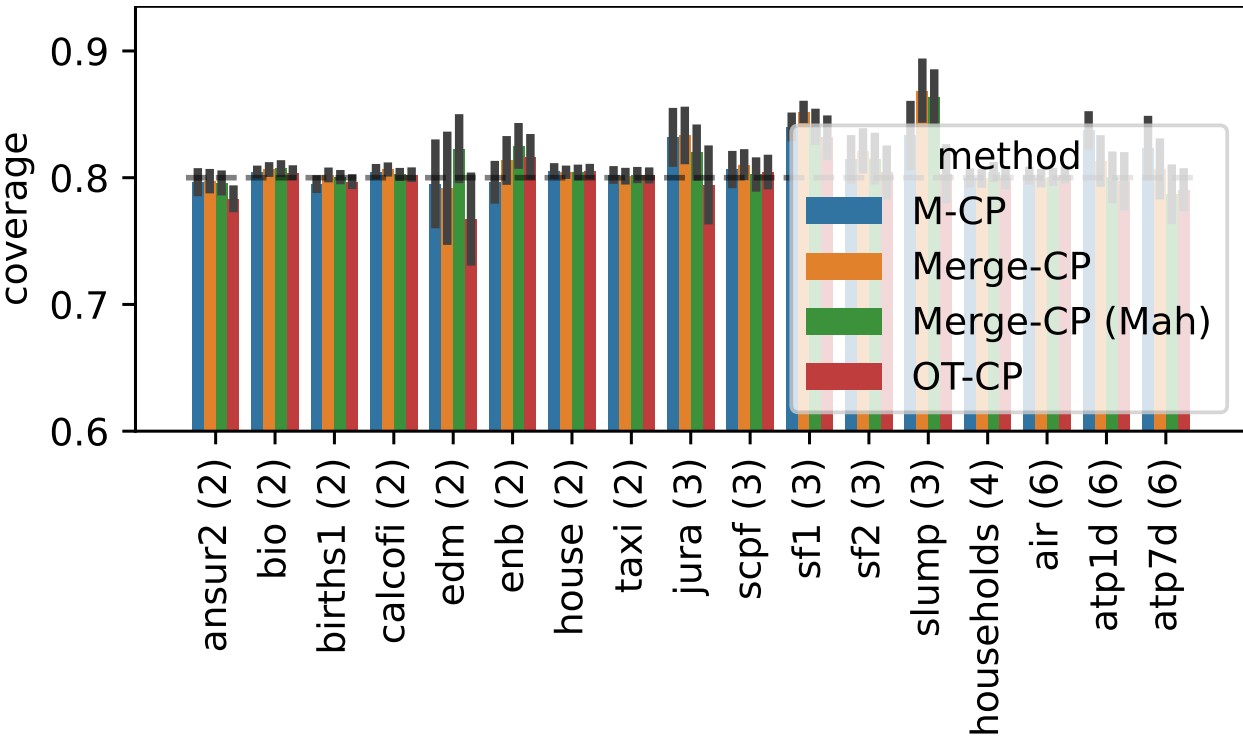

Figure 11: Coverage of all baselines on small dimensional datasets, corresponding to the region sizes given in Figure 5a.

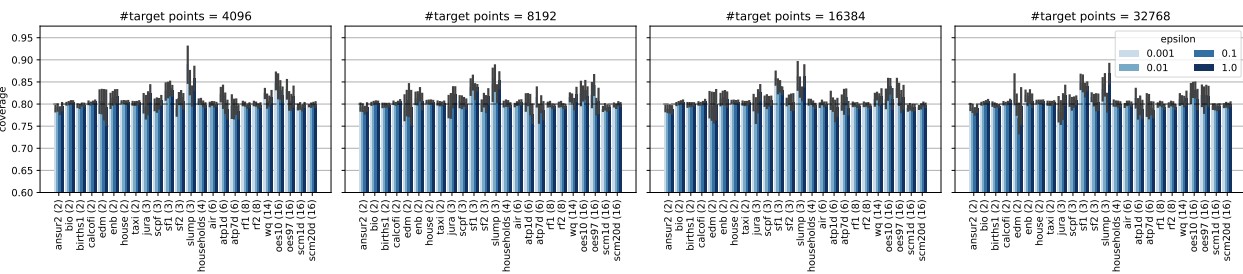

Figure 12: Ablation: coverage quality as a function of hyperparameters, with the setting corresponding to Figure 7.

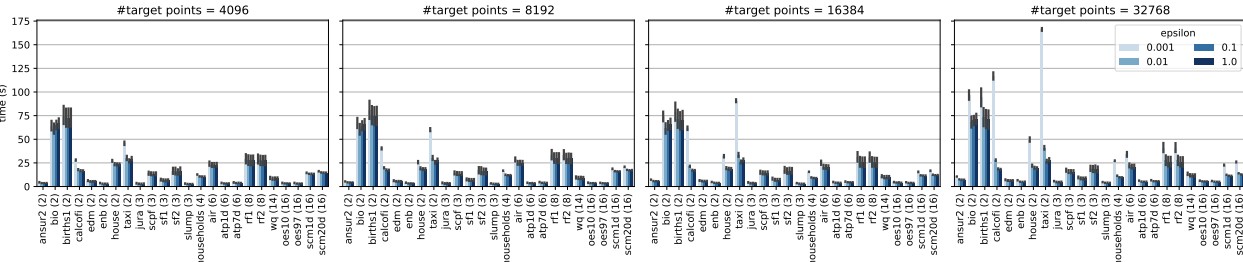

Figure 13: Ablation: running time as a function of hyperparameters, with the setting corresponding to Figure 7 .

| $\varepsilon$ | #target | ansur2 (2) | bio (2) | births1 (2) | calcofi (2) | edm (2) | enb (2) | house (2) | taxi (2) | jura (3) | scpf (3) | sf1 (3) | sf2 (3) |
|---|---|---|---|---|---|---|---|---|---|---|---|---|---|
| 0.001 | 4096 | $3.3 \pm 0.064$ | $0.46 \pm 0.057$ | $78 \pm 70$ | $2.6 \pm 0.089$ | $1.9 \pm 0.3$ | $0.81 \pm 0.21$ | $2 \pm 0.051$ | $7 \pm 0.12$ | $13 \pm 2.6$ | $0.78 \pm 0.4$ | $14 \pm 2.6$ | $0.82 \pm 0.32$ |
| | 8192 | $3.4 \pm 0.059$ | $0.45 \pm 0.057$ | $78 \pm 70$ | $2.6 \pm 0.089$ | $1.9 \pm 0.29$ | $0.81 \pm 0.2$ | $2 \pm 0.05$ | $7 \pm 0.13$ | $11 \pm 2.6$ | $0.73 \pm 0.23$ | $16 \pm 3.9$ | $0.4 \pm 0.16$ |
| | 16384 | $3.4 \pm 0.059$ | $0.46 \pm 0.058$ | $78 \pm 70$ | $2.6 \pm 0.093$ | $1.8 \pm 0.28$ | $0.83 \pm 0.21$ | $2 \pm 0.048$ | $7 \pm 0.13$ | $12 \pm 2.3$ | $0.87 \pm 0.34$ | $21 \pm 4.8$ | $0.44 \pm 0.2$ |
| | 32768 | $3.4 \pm 0.063$ | $0.46 \pm 0.058$ | $78 \pm 70$ | $2.6 \pm 0.092$ | $1.9 \pm 0.3$ | $0.81 \pm 0.2$ | $2 \pm 0.05$ | $7 \pm 0.13$ | $12 \pm 2.6$ | $1.2 \pm 0.47$ | $16 \pm 2.9$ | $0.57 \pm 0.18$ |
| 0.01 | 4096 | $3.3 \pm 0.055$ | $0.55 \pm 0.12$ | $78 \pm 70$ | $2.5 \pm 0.084$ | $1.9 \pm 0.3$ | $0.81 \pm 0.21$ | $2 \pm 0.05$ | $7.5 \pm 0.63$ | $11 \pm 2.8$ | $0.43 \pm 0.15$ | $12 \pm 2.1$ | $0.2 \pm 0.086$ |
| | 8192 | $3.3 \pm 0.054$ | $0.56 \pm 0.13$ | $78 \pm 70$ | $2.5 \pm 0.082$ | $1.8 \pm 0.3$ | $0.8 \pm 0.21$ | $2 \pm 0.049$ | $7.5 \pm 0.69$ | $10 \pm 2.6$ | $0.37 \pm 0.15$ | $12 \pm 2.8$ | $0.17 \pm 0.063$ |
| | 16384 | $3.3 \pm 0.045$ | $0.56 \pm 0.12$ | $78 \pm 70$ | $2.5 \pm 0.082$ | $1.7 \pm 0.24$ | $0.8 \pm 0.21$ | $2 \pm 0.05$ | $7.5 \pm 0.71$ | $13 \pm 4.3$ | $0.4 \pm 0.18$ | $11 \pm 2.9$ | $0.19 \pm 0.076$ |
| | 32768 | $3.3 \pm 0.064$ | $0.56 \pm 0.12$ | $78 \pm 70$ | $2.5 \pm 0.085$ | $1.7 \pm 0.26$ | $0.82 \pm 0.22$ | $2 \pm 0.049$ | $7.5 \pm 0.69$ | $10 \pm 2.7$ | $0.41 \pm 0.17$ | $12 \pm 2.6$ | $0.18 \pm 0.071$ |
| 0.1 | 4096 | $3.3 \pm 0.058$ | $0.49 \pm 0.011$ | $78 \pm 70$ | $2.5 \pm 0.084$ | $1.6 \pm 0.25$ | $0.81 \pm 0.21$ | $2.3 \pm 0.065$ | $8.3 \pm 1.4$ | $9.2 \pm 2.8$ | $0.37 \pm 0.15$ | $6.6 \pm 0.96$ | $0.48 \pm 0.1$ |
| | 8192 | $3.3 \pm 0.059$ | $0.49 \pm 0.011$ | $78 \pm 70$ | $2.5 \pm 0.084$ | $1.6 \pm 0.26$ | $0.8 \pm 0.21$ | $2.3 \pm 0.065$ | $8.2 \pm 1.5$ | $9.4 \pm 2.9$ | $0.4 \pm 0.15$ | $6.1 \pm 0.89$ | $0.53 \pm 0.11$ |
| | 16384 | $3.3 \pm 0.054$ | $0.49 \pm 0.012$ | $78 \pm 70$ | $2.5 \pm 0.081$ | $1.6 \pm 0.26$ | $0.8 \pm 0.21$ | $2.3 \pm 0.058$ | $8.2 \pm 1.4$ | $9.4 \pm 2.9$ | $0.37 \pm 0.12$ | $6.4 \pm 0.83$ | $0.45 \pm 0.092$ |
| | 32768 | $3.3 \pm 0.051$ | $0.49 \pm 0.011$ | $77 \pm 70$ | $2.5 \pm 0.083$ | $1.5 \pm 0.25$ | $0.79 \pm 0.2$ | $2.3 \pm 0.057$ | $8.2 \pm 1.4$ | $8.9 \pm 2.9$ | $0.36 \pm 0.12$ | $6.5 \pm 1.2$ | $0.5 \pm 0.1$ |
| 1 | 4096 | $3.6 \pm 0.055$ | $0.65 \pm 0.019$ | $78 \pm 70$ | $2.5 \pm 0.1$ | $1.7 \pm 0.27$ | $0.92 \pm 0.24$ | $3 \pm 0.13$ | $6.4 \pm 0.14$ | $13 \pm 4$ | $0.45 \pm 0.16$ | $9.5 \pm 1.9$ | $0.84 \pm 0.13$ |
| | 8192 | $3.6 \pm 0.067$ | $0.59 \pm 0.013$ | $78 \pm 70$ | $2.5 \pm 0.099$ | $1.7 \pm 0.26$ | $0.91 \pm 0.24$ | $3 \pm 0.14$ | $6.3 \pm 0.14$ | $13 \pm 4$ | $0.42 \pm 0.14$ | $10 \pm 1.8$ | $0.93 \pm 0.16$ |
| | 16384 | $3.5 \pm 0.072$ | $0.57 \pm 0.016$ | $78 \pm 70$ | $2.5 \pm 0.099$ | $1.7 \pm 0.27$ | $0.91 \pm 0.24$ | $3 \pm 0.13$ | $6.4 \pm 0.14$ | $14 \pm 4$ | $0.48 \pm 0.17$ | $9.8 \pm 1.7$ | $0.91 \pm 0.17$ |
| | 32768 | $3.5 \pm 0.061$ | $0.6 \pm 0.028$ | $78 \pm 71$ | $2.5 \pm 0.1$ | $1.7 \pm 0.27$ | $0.91 \pm 0.24$ | $2.9 \pm 0.13$ | $6.4 \pm 0.15$ | $13 \pm 4$ | $0.47 \pm 0.17$ | $10 \pm 1.7$ | $0.9 \pm 0.17$ |

Table 2: Mean region size for varying $\varepsilon$ and the number of target points in the ball.

| $\varepsilon$ | #target | slump (3) | households (4) | air (6) | atp1d (6) | atp7d (6) |
|---|---|---|---|---|---|---|
| 0.001 | 4096 | $15 \pm 7.6$ | $37 \pm 1.4$ | $2.6 \times 10^3 \pm 1.9 \times 10^3$ | $81 \pm 19$ | $8.5 \times 10^2 \pm 4.5 \times 10^2$ |
| | 8192 | $7.9 \pm 2$ | $36 \pm 1.9$ | $7.1 \times 10^2 \pm 56$ | $99 \pm 41$ | $5.9 \times 10^2 \pm 1.8 \times 10^2$ |
| | 16384 | $11 \pm 3.7$ | $34 \pm 1.3$ | $6.9 \times 10^2 \pm 52$ | $65 \pm 19$ | $9.4 \times 10^2 \pm 3 \times 10^2$ |
| | 32768 | $12 \pm 4.3$ | $36 \pm 2.6$ | $6.8 \times 10^2 \pm 36$ | $87 \pm 28$ | $5.1 \times 10^2 \pm 2 \times 10^2$ |
| 0.01 | 4096 | $20 \pm 6.8$ | $37 \pm 1.6$ | $8.5 \times 10^2 \pm 1 \times 10^2$ | $85 \pm 24$ | $7.9 \times 10^2 \pm 4.1 \times 10^2$ |
| | 8192 | $12 \pm 4.9$ | $34 \pm 1.7$ | $1.3 \times 10^3 \pm 7 \times 10^2$ | $82 \pm 24$ | $4 \times 10^2 \pm 1.5 \times 10^2$ |
| | 16384 | $7.1 \pm 2.2$ | $33 \pm 0.81$ | $5.5 \times 10^2 \pm 47$ | $1.1 \times 10^2 \pm 26$ | $3.7 \times 10^2 \pm 68$ |
| | 32768 | $10 \pm 4$ | $31 \pm 0.97$ | $4.8 \times 10^2 \pm 51$ | $42 \pm 9.1$ | $2.8 \times 10^2 \pm 98$ |
| 0.1 | 4096 | $5.8 \pm 1.3$ | $27 \pm 1.3$ | $3.2 \times 10^2 \pm 32$ | $8.1 \pm 1.7$ | $33 \pm 9.2$ |
| | 8192 | $5.9 \pm 1.3$ | $26 \pm 1.3$ | $3.1 \times 10^2 \pm 33$ | $5.7 \pm 1$ | $27 \pm 6.9$ |
| | 16384 | $5.9 \pm 1.4$ | $25 \pm 1$ | $3.1 \times 10^2 \pm 34$ | $4 \pm 1.4$ | $26 \pm 7.7$ |
| | 32768 | $5.1 \pm 1.1$ | $25 \pm 1$ | $3.1 \times 10^2 \pm 34$ | $3.8 \pm 0.88$ | $16 \pm 5.1$ |
| 1 | 4096 | $14 \pm 5.3$ | $29 \pm 1.3$ | $4.3 \times 10^2 \pm 31$ | $6.2 \pm 1.7$ | $69 \pm 25$ |
| | 8192 | $15 \pm 5.3$ | $30 \pm 2.1$ | $3.4 \times 10^2 \pm 38$ | $5.6 \pm 2.2$ | $69 \pm 25$ |
| | 16384 | $16 \pm 5.6$ | $28 \pm 1.1$ | $4.1 \times 10^2 \pm 36$ | $6.1 \pm 2$ | $76 \pm 27$ |
| | 32768 | $15 \pm 5.5$ | $29 \pm 1.9$ | $4.3 \times 10^2 \pm 38$ | $5.6 \pm 1.5$ | $73 \pm 24$ |

Table 3: Mean region size for varying $\varepsilon$ and the number of target points in the ball.

| $\varepsilon$ | #target | rf1 (8) | rf2 (8) | wq (14) | oes10 (16) | oes97 (16) | scm1d (16) | scm20d (16) |
|---|---|---|---|---|---|---|---|---|
| 0.001 | 4096 | $2 \times 10^{13} \pm 2 \times 10^{13}$ | $2 \times 10^{13} \pm 2 \times 10^{13}$ | $7.1 \times 10^9 \pm 3 \times 10^9$ | $2.9 \times 10^8 \pm 8.3 \times 10^7$ | $8.7 \times 10^8 \pm 4 \times 10^8$ | $4 \times 10^7 \pm 3.6 \times 10^7$ | $1.7 \times 10^7 \pm 1.1 \times 10^7$ |
| | 8192 | $2 \times 10^{13} \pm 2 \times 10^{13}$ | $2 \times 10^{13} \pm 2 \times 10^{13}$ | $3.7 \times 10^9 \pm 1.9 \times 10^9$ | $3.7 \times 10^8 \pm 1.3 \times 10^8$ | $1.4 \times 10^9 \pm 1.2 \times 10^9$ | $9.3 \times 10^5 \pm 5 \times 10^5$ | $2.5 \times 10^8 \pm 1.9 \times 10^8$ |
| | 16384 | $2 \times 10^{13} \pm 2 \times 10^{13}$ | $2 \times 10^{13} \pm 2 \times 10^{13}$ | $6.6 \times 10^9 \pm 3.2 \times 10^9$ | $5.6 \times 10^8 \pm 4.3 \times 10^8$ | $2.5 \times 10^8 \pm 1.3 \times 10^8$ | $3.5 \times 10^5 \pm 1.3 \times 10^5$ | $8.9 \times 10^7 \pm 5.7 \times 10^7$ |
| | 32768 | $2 \times 10^{13} \pm 2 \times 10^{13}$ | $2 \times 10^{13} \pm 2 \times 10^{13}$ | $3.1 \times 10^9 \pm 1.2 \times 10^9$ | $5.5 \times 10^8 \pm 3 \times 10^8$ | | $9.7 \times 10^5 \pm 4.5 \times 10^5$ | $1.3 \times 10^9 \pm 1.3 \times 10^9$ |
| 0.01 | 4096 | $2 \times 10^{13} \pm 2 \times 10^{13}$ | $2 \times 10^{13} \pm 2 \times 10^{13}$ | $1.1 \times 10^{10} \pm 7.3 \times 10^9$ | $4.3 \times 10^9 \pm 3.8 \times 10^9$ | $3.5 \times 10^9 \pm 2.5 \times 10^9$ | $4.1 \times 10^8 \pm 3.8 \times 10^8$ | $1.3 \times 10^{11} \pm 1.1 \times 10^{11}$ |
| | 8192 | $2 \times 10^{13} \pm 2 \times 10^{13}$ | $2 \times 10^{13} \pm 2 \times 10^{13}$ | $6.4 \times 10^{10} \pm 6 \times 10^{10}$ | $3 \times 10^{10} \pm 2.8 \times 10^{10}$ | $1 \times 10^{10} \pm 6.1 \times 10^9$ | $8.1 \times 10^8 \pm 5.5 \times 10^8$ | $1.1 \times 10^{11} \pm 1.1 \times 10^{11}$ |
| | 16384 | $2 \times 10^{13} \pm 2 \times 10^{13}$ | $2 \times 10^{13} \pm 2 \times 10^{13}$ | $1.1 \times 10^9 \pm 7.9 \times 10^8$ | $1.1 \times 10^9 \pm 4.3 \times 10^8$ | $1 \times 10^{10} \pm 5.7 \times 10^9$ | $4.8 \times 10^7 \pm 3.7 \times 10^7$ | $1.3 \times 10^9 \pm 8.3 \times 10^8$ |
| | 32768 | $2 \times 10^{13} \pm 2 \times 10^{13}$ | $2 \times 10^{13} \pm 2 \times 10^{13}$ | $5.1 \times 10^{11} \pm 4.9 \times 10^{11}$ | $6.5 \times 10^9 \pm 5 \times 10^9$ | $4 \times 10^9 \pm 3.2 \times 10^9$ | $1.6 \times 10^7 \pm 9.5 \times 10^6$ | $2.7 \times 10^8 \pm 1.3 \times 10^8$ |
| 0.1 | 4096 | $2 \times 10^{13} \pm 2 \times 10^{13}$ | $2 \times 10^{13} \pm 2 \times 10^{13}$ | $8.7 \times 10^9 \pm 3.7 \times 10^9$ | $4.8 \times 10^4 \pm 3.2 \times 10^4$ | $6 \times 10^9 \pm 6 \times 10^9$ | $1.5 \times 10^3 \pm 6.7 \times 10^2$ | $1.3 \times 10^6 \pm 6.4 \times 10^5$ |
| | 8192 | $2 \times 10^{13} \pm 2 \times 10^{13}$ | $2 \times 10^{13} \pm 2 \times 10^{13}$ | | $1.7 \times 10^5 \pm 1.3 \times 10^5$ | $6 \times 10^9 \pm 6 \times 10^9$ | $6.2 \times 10^2 \pm 2.8 \times 10^2$ | $1.2 \times 10^6 \pm 8.7 \times 10^5$ |
| | 16384 | $2 \times 10^{13} \pm 2 \times 10^{13}$ | $2 \times 10^{13} \pm 2 \times 10^{13}$ | $1.3 \times 10^{10} \pm 6.8 \times 10^9$ | $5.2 \times 10^4 \pm 4.7 \times 10^4$ | $5.6 \times 10^9 \pm 5.6 \times 10^9$ | $2.2 \times 10^2 \pm 46$ | $2.9 \times 10^5 \pm 1 \times 10^5$ |
| | 32768 | $2 \times 10^{13} \pm 2 \times 10^{13}$ | $2 \times 10^{13} \pm 2 \times 10^{13}$ | $7.4 \times 10^9 \pm 2.9 \times 10^9$ | $7.6 \times 10^3 \pm 5.1 \times 10^3$ | $9.2 \times 10^7 \pm 8.1 \times 10^7$ | $1.1 \times 10^2 \pm 17$ | $1.1 \times 10^5 \pm 3.1 \times 10^4$ |
| 1 | 4096 | $2 \times 10^{13} \pm 2 \times 10^{13}$ | $2 \times 10^{13} \pm 2 \times 10^{13}$ | $8 \times 10^8 \pm 2 \times 10^8$ | $6.6 \times 10^2 \pm 3.4 \times 10^2$ | $8.3 \times 10^5 \pm 8.1 \times 10^5$ | $4.1 \times 10^2 \pm 76$ | $5.2 \times 10^5 \pm 6.5 \times 10^4$ |
| | 8192 | $2 \times 10^{13} \pm 2 \times 10^{13}$ | $2 \times 10^{13} \pm 2 \times 10^{13}$ | $6.9 \times 10^8 \pm 1.7 \times 10^8$ | $3.5 \times 10^2 \pm 1.8 \times 10^2$ | $7.7 \times 10^5 \pm 7.6 \times 10^5$ | $8.5 \times 10^2 \pm 3.1 \times 10^2$ | $1.1 \times 10^6 \pm 3.9 \times 10^5$ |
| | 16384 | $2 \times 10^{13} \pm 2 \times 10^{13}$ | $2 \times 10^{13} \pm 2 \times 10^{13}$ | $5.3 \times 10^8 \pm 1.2 \times 10^8$ | $2.2 \times 10^2 \pm 1.5 \times 10^2$ | $4 \times 10^5 \pm 4 \times 10^5$ | $1.3 \times 10^2 \pm 14$ | $4.7 \times 10^5 \pm 1.8 \times 10^5$ |
| | 32768 | $2 \times 10^{13} \pm 2 \times 10^{13}$ | $2 \times 10^{13} \pm 2 \times 10^{13}$ | $5.5 \times 10^8 \pm 1.5 \times 10^8$ | $1.9 \times 10^2 \pm 1.6 \times 10^2$ | $3.1 \times 10^5 \pm 3.1 \times 10^5$ | $1 \times 10^2 \pm 11$ | $3.4 \times 10^5 \pm 6.4 \times 10^4$ |

Table 4: Mean region size for varying $\varepsilon$ and the number of target points in the ball.

