# OpenReview forum: "Multivariate Conformal Prediction using Optimal Transport"
_TMLR — Accepted by TMLR_

### Review · Reviewer_Hpa6 · 2025-12-01

**Summary Of Contributions:**

The paper presents a method to produce geometry-aware nonconformal scores for a multivariate response using tools from Optimal Transport. The idea is to transport the empirical distribution of $\mathbb{R}^d$-valued quantile nonconformal scores towards a spherical uniform distribution in $\mathbb{R}^d$. The paper then leverages the definition of spherical rank by Chernozhukov et al (2017), given by the norm of transported vectors, as a new scalar nonconformity score that can be use in conformal prediction tasks. The paper proposes an entropic approximation of such transportation map depending on a parameter $\epsilon>0$ that controls softness and guarantees a unique solution. The procedure guarantees marginal coverage, but the paper also presents a methodological approach for approximate conditional coverage, incurring computational trade-off.

**Audience:**

Yes

**Audience Explanation:**

Yes, I believe some of TMLR’s readership would be interested in the paper’s findings. However, in their current form the results are mostly incremental relative to the existing literature and concurrent work. The paper would provide substantially greater value to readers if its theoretical and methodological contributions were strengthened.

**Broader Impact Concerns:**

None identified.

**Claims And Evidence:**

Yes

**Claims Explanation:**

The paper presents its main claims in simpler terms, which facilitates readability. Yet, while the claims are clear, a more detailed theoretical justification is missing. Although the paper does not offer new theoretical contributions and therefore does not introduce theorems or remarks requiring formal proofs, the theoretical discussion should be more substantial. In particular, the need for a geometry-aware solution requires a stronger justification that clearly differentiates the approach from Merge-CP. If the central claims rest on the importance of data geometry, that theoretical perspective should be developed more fully.

Empirically, the proposed method’s marginal coverage is not markedly better than that of Merge-CP, which is a much simpler off-the-shelf baseline. This places the paper in a weak position, since the empirical improvements in marginal coverage are small and the theoretical motivation is underdeveloped. There are modest gains in WSC and region size on some benchmarks, but these gains are not consistent, likely because performance depends on the geometry of each dataset. Approximate conditional coverage is more competitive on some datasets, presumably those with a rich data topology that the OT component captures well.

**Requested Changes:**

Given that the paper’s contribution is incremental relative to the existing literature and concurrent work, I strongly recommend that the authors strengthen the theoretical and methodological components. In particular, discussion of the following points would help:

- Beyond the need to collapse vector-valued nonconformity scores to scalars, the geometry and topology of the data should be the main rationale for using OT within CP, making it theoretically more justified than Merge-CP. The paper briefly mentions skewed feature-data distributions but does not develop geometry as a central argument. Some variability in the benchmark results likely reflects differences in dataset geometry. I recommend deepening the justification along these lines.
- The method relies entirely on the spherical rank of Chernozhukov et al. (2017). Which properties of this construction make the resulting real-valued nonconformity scores attractive? For instance, the induced score is bounded by design, unlike many scores in the scalar setting. How does this boundedness help or hinder CP tasks? Since Chernozhukov et al. (2017) discuss other possible target distributions, I recommend analyzing the theoretical implications of the specific choices made here.
- The entropic map is presented in dual form. While memory-efficient, the dual can be less stable for very small $\epsilon$ than primal Sinkhorn. Consider discussing trade-offs with the primal formulation, and discussing the case use of accelerated variants such as Greenkhorn, Anderson acceleration, or Sinkhorn–Newton.

### Recommended style changes

- **Citation formatting**: There are inconsistencies between textual and parenthetical citations. For example, the opening sentence _"Conformal prediction (CP) Gammerman et al. (1998); Vovk et al. (2005); Shafer \& Vovk (2008) has emerged…"_ should read _"Conformal prediction (CP) (Gammerman et al., 1998; Vovk et al., 2005; Shafer \& Vovk, 2008) has emerged…"_.  Likewise, _"As recalled in (Dheur et al., 2025)…"_  should be _"As recalled by Dheur et al. (2025)…"_.
-  **Typos and confusing text**: For instance, _"We are interested in setting where…"_ should rather be _"We are interested in settings where…"_ (or _"We are interested in the setting where…"_).
- **Clarity and pacing**: Several ideas are presented too quickly, which reduces clarity. Consider adding brief transitions, precise definitions, or small examples to improve readability.

---

> ### Author Response · Authors · 2026-03-07
>
> We thank the reviewer for the thoughtful comments on geometric motivation, spherical ranks, numerical aspects, and presentation.
>
> > Geometry / topology motivation relative to Merge-CP
>
> We strengthened this point substantially in the revised manuscript.
> The introduction and Section 3.1 now explain more clearly that fixed norm-based scalarizations induce ellipsoidal level sets, whereas OTCP can adapt to non-elliptical empirical score geometry through the transport map and the preimage of spherical shells. This was intended to clarify that OT is not introduced merely as a way to collapse vector scores, but as a geometry-aware scalarization.
> We agree that this point is mainly strengthened at the level of conceptual motivation and exposition; the current revision does not add a new diagnostic analysis explicitly predicting when OTCP should outperform Merge-CP.
>
> > Marginal coverage versus empirical comparisons
>
> We revised the results narrative so that marginal coverage is treated primarily as a validity check, while efficiency and conditional-coverage-oriented metrics are emphasized more clearly as the more informative empirical comparisons.
>
> > Spherical-rank properties and alternative references
>
> This discussion is now much clearer in Section 2.3. The revised manuscript explains that under the spherical reference:
> - the transported radius acts as a scalar multivariate rank,
> - the normalized direction acts as a multivariate sign, and the geometry of the induced depth / quantile regions depends on the chosen reference distribution.
> We also expanded the explanation of why the spherical reference is a natural center-outward benchmark in our setting.
>
> > Dual entropic map stability for small $\varepsilon$
> We added a short numerical discussion in Section 2.4 noting that very small $\varepsilon$ can lead to numerically stiff Sinkhorn iterations, and we now mention stabilized / primal formulations and standard accelerations from the OT literature.
> We did not experimentally evaluate those alternatives in this paper, but the revision now acknowledges that design space more explicitly.
>
> > Style changes
> We carried out a broad style pass across the manuscript. The current revision is cleaner in wording, notation, and citations than the original version, and we will continue refining local presentation in the final version.
>
> We thank the reviewer again for the helpful comments, which directly improved both the motivation and the numerical discussion in the revised paper.

---

### Review · Reviewer_bWf2 · 2026-01-05

**Summary Of Contributions:**

The submission considers conformal prediction based on multivariate scores---the latter arise naturally when predicting multivariate observations. In univariate conformal prediction, scores calculated on the calibration set are ranked and empirical quantiles are given by these order statistics. However, there is no canonical order on multivariate scores. Previous approaches are basically transforming multivariate scores into univariate scores in some supposedly natural way preserving geometry, intuitions, etc.; for instance, by taking norms under some well-chosen symmetric definite positive matrix (ideally, the unknown covariance matrix of the scores).

The key idea of this submission is to resort to the theory of optimal transport for this multivariate-to-univariate conversion, which could then be considered being optimally performed. The submission details in particular how to implement, in a computationally efficient way, this (approximate) optimal-transport conversion. This key idea and its developments form the main strength of the submission (but see however * below)

The theoretical results are about coverage (this kind of results follows directly from the Vovk et al., 2005 alma-matter result of this theory). Simulations cover more aspects: coverage, efficiency (i.e., sizes of the confidence regions) and computation times, on a many data sets of varied sizes.

[*] One important point to mention, at this stage, is that a simultaneous and concurrent work exists, as acknowledged by the submission: one ICML'25 article by Thurin et al. The submission mentions that both were written and put on arXiv at the same time: I let the associate editor perform this check, that I on purpose did not operate because of the double-blind policy. I did however quickly scan through the Thurin et al. (2025) submission, and indeed, the key idea is similar, but, as mentioned by this submission, there exist differences in the approaches. In a nutshell, this submission covers better the computational aspects and provides more extensive simulations. *If* (and only if) the simultaneous arXiv submission claim is true, I would not see the existence of this other paper as a main issue for publication of this submission at TMLR.

[**] Disclaimer: I am not specialist of optimal transport, but I did contribute to multivariate conformal prediction. As the rest of the report will reveal, I had difficulties in following the exposition (and I do not know whether this is due, or not, to my lack of familiarity with optimal transport). The main (and perhaps only) weakness of this submission lies  therefore in the exposition.

**Additional Comments:**

The way how references are cited should be corrected throughout the document---they do not naturally fit in the sentences (they should often be provided in parentheses or be introduced with a '; see')

In Section 2.2, consider also citing the works by Messoudi et al.

Top of page 4: the text says 'the uniform ball on $B(0,1)$' but how is the latter defined? Later text (and in particular, the fact that the norm is uniformly distributed over $[0,1]$) rather hints to a distribution of the form: first pick the norm uniformly at random and then, pick an angle uniformly at random, which is different from the uniform distribution over the ball, given by the normalized Lebesgue thereon.

Section 2.4, first line thereof: indicate (I guess) that the $z_j$ are distributed according to $\mathbb{P}$ and the $u_k$ according to $\mathbb{U}$; also, I wonder: OK, one ignores $\mathbb{P}$  and is happy to only see samples thereof, but why would one have to sample some $u_k$'s according to $\mathbb{U}$, which is perfectly known?

Section 3.2: Does this method generalize or does it have links with Eq. (6) --- e.g., are exponential weights replaced by some kernel $H$?

Typos:
- $x_{n+1} \to x$ on page 2, line -11
- In Section 2.4, the ${\boldsymbol{f}_i^\star}$ and ${\boldsymbol{g}^\star_j}$ shoud not be bold: $f^\star_i$ and $g^\star_f$, following what I perceived where the conventions used in the submission

**Audience:**

Yes

**Audience Explanation:**

The topic is timely and of practical use, the key idea of a multivariate-to-univariate conversion of scores through optimal transport is interesting and open new research avenues, i.e., there is absolutely no concern about the submission being of interest for the TMLR community.

**Claims And Evidence:**

Yes

**Claims Explanation:**

Yes, they are, though some parts thereof are improvable
- The theory claims about coverage are OK (though they could be written in a more formal way, see the section *Requested Changes* below)
- The simulations are extensive indeed
- The comparison to existing work should be improved and clarified, on various grounds, and I detail this now.

To me, despite the claimed simultaneous obtention of results, the concurrent article by Thurin et al. is already published, so it would make sense to provide more detailed comparisons along the text and not just in a final short section. Also, there are several parts of the comparison that I could not follow.

For the claim that this submission does not compromise finite-sample guarantees: I guess this hints at the results of Section 3.1. Therein, after formally stating the results, there could be a remark that would first summarize the nature of the results by Thurin et al. concerning coverage guarantees, and then, a discussion of why the results of the present submission are more satisfactory.

The same holds for the claim about the improved computational performance, with a remark to be written in Section 3.3 and probably also a comment much earlier, at some point in Section 2, indicating that one of the key distinguishing features of this submission is to deal with the computational aspects of implementing optimal transport.

Finally, the comment about the breadth and extent of the simulations would be the last comment of Section 6: it could rather be included in Section 4, together with a summary of the kind of data sets Thurin et al. consider. It is worth noting that the latter reference explicitly mentions that this submission offers a better computational solution and that it is suited for the large-scale scenario.

**Requested Changes:**

My main requests (on top of detailing more the comparison to the concurrent work, see above) would concern the exposition.

First, following Section 2.2, it would be worth noting (perhaps as a Corollary of Proposition 2.1) a general procedure, with a general coverage guarantee (formally stated and proved, along the lines of the top of page 5). The procedure would cut data into: train data, hold data, calibration data, and test data, where hold data is used to estimate some multivariate-to-univariate conversion function. That function could be an approximate entropic map, or an empirically estimated covariance matrix (for the Mahalanobis-ellipsoid approaches). Doing so, this submission would put into perspective existing techniques and unify their exposition better. The submission would then proceed by explaining how the multivariate-to-univariate conversion function is built.

In particular, the last paragraph of Section 3.1 could be expanded and stated right after this general procedure.

Concerning the reminders on optimal transport: I would suggest to put them in a separate section (that readers familiar with it would skip). They are of two kinds: some general exposition (the top of page 4 is remarkably pedagogical) and a better motivation and a description of the computational aspects (readers unfamiliar with the topic wonder what the transition from Section 2.3 to Section 2.4 is).

In Section 3.1, I had difficulties to identify whether the submission wants to consider $T^\star$ or $T_{\epsilon}$. To me, the conversion, quantiles, confidence regions are all built with $T_\epsilon$. But the text mentions on line 6 of page 5 that quantiles are computed with $S_{\mbox{OT-CP}}$ which is defined through $T^\star$ (see line -4 of page 4). Also, the text mentions some $\hat{T}$ on line 7 of page 7, and I didn't find the definition thereof. All in all, this part of the submission would eventually be summarized in one line, saying that $T_{\epsilon}$ + taking norms is considered as the conversion function, following the general scheme that will have been introduced earlier.

Finally, I am unsure about the aims and methods of Section 3.2. It might be useful to formally introduce first the concept of conditional coverage, explain that other targets are now considered than plain conformal prediction, etc. The transition from Section 3.1 to Section 3.2 is a bit rough. The argument on top of page 6 looks like a general argument, so perhaps the same idea of a general procedure, for which coverage guarantees are provided through some general argument, could be replicated here.

---

> ### Author Response · Authors · 2026-03-07
>
> We thank the reviewer for the careful reading and especially for the detailed suggestions on exposition, conditional coverage, and the comparison to concurrent work.
>
> > Comparison to Thurin et al. (2025)
>
> We agree that this comparison must be stated carefully.
> We strengthened it in the revised manuscript across the main paper
> Section 3.1 now contrasts more clearly:
> - their use of MK ranks computed by discrete assignment on the calibration set,
> - our use of an entropic map learned on an independent hold-out split and then reused out of sample.
>
> We also clarified the main computational distinction: their formulation is assignment-based with effectively $n=m$, whereas our entropic-map implementation scales as $O(nm)$, allows $n\neq m$, and yields an explicit reusable map.
>
> > Unified train / hold / calibration pipeline
>
> We addressed the main conceptual point raised in the review.
> In particular, we now make explicit that the validity mechanism is not OT-specific: once a scalarization is learned on data independent of the calibration set, standard split conformal inference applies. The revised manuscript reflects this in two places:
>
> -  Section 2.2.1 now introduces a general scalarization template,
> - Section 3.1 now presents the explicit train / hold / calibration split and explains the resulting validity mechanism.
> We agree that a boxed “general split conformal with learned scalarization” template could make this even clearer, but the current revision already addresses the core substance of the reviewer’s suggestion.
>
> > OT background and the Section 2.3 \rightarrow 2.4 transition
> We revised this transition by expanding the explanation of spherical ranks and by making the role of the entropic map estimator more explicit in Section 2.4.
> We agree that this part remains mathematically dense, but we believe it is now substantially clearer than in the previous draft.
>
> > Notational ambiguity in Section 3.1
> This was one of the main targets of the revision. Section 3.1 now distinguishes more clearly between the population scalarization and the practical entropic-map scalarization, and uses the latter consistently in the split-conformal procedure and prediction-region definition.
>
> > Conditional coverage in Section 3.2
>
> We substantially revised Section 3.2 in response to the review.
> The revised manuscript now:
> - introduces object-conditional validity and recalls why exact distribution-free object-conditional validity is impossible in general,
> - states an exact finite-sample cell-conditional validity result for the hard-partition setting,
>  and distinguishes this exact result from soft localization, which is now explicitly presented as heuristic and without a formal distribution-free finite-sample guarantee.
> This was one of the most important clarifications in the revision.
>
> > Reference distribution on the unit ball
>
> We clarified the spherical reference distribution more explicitly in Section 2.3: uniform direction on the sphere together with an independent uniform radius on [0,1], along with the corresponding rank / sign interpretation.
>
> > Clarification of source vs. target sampling in Section 2.4
>
> This is also clarified in the revision. Section 2.4 now states more explicitly that the z_i are source score samples, while the u_j are reference points from the known spherical target used to discretize the OT problem.
>
> > Kernel localization vs. Gibbs / Sinkhorn weights
> We now clarify that input-space kernel weighting modifies the empirical source distribution used to estimate local transport maps, and is distinct from the Gibbs / Sinkhorn weights appearing inside the entropic OT map itself.
>
> > Citation style and typos
>
> We revised citations and corrected many style and formatting issues throughout the manuscript. We believe the main presentation concerns raised in the review are addressed in the current revision, though we will continue polishing the final version.
>
> We thank the reviewer again for the constructive suggestions, which were especially helpful in improving the paper’s structure and the precision of the validity discussion.

---

> > ### Comment · Reviewer_bWf2 · 2026-03-08
> > **Merge end of Section 3.1 and Section 6**
> >
> > Thanks for the revised PDF and the answers --- I am generally fine with them.
> >
> > However, I would strongly recommend merging the two parts where the concurrent work by Thurin et al. is discussed, namely the end of Section 3.1 and Section 6 --- rather in Section 3.1, with a pointer very early and very clearly in the paper that this is concurrent work.

---

### Review · Reviewer_No3E · 2026-02-21

**Summary Of Contributions:**

The authors propose a new non-conformity score for conformal prediction with multivariate response variables. They use entropic maps, fit on a hold-out set, to transform the element-wise (vector) CQR scores into the spherical uniform reference measure. The norms of the transformed scores are approximately uniform and then enter conformal calibration via the usual empirical ranking, which preserves distribution-free validity in finite samples. Approximate conditional coverage is demonstrated by localizing the transport map in each partition of the input space.

Empirically, there is moderate improvement relative to baselines for datasets of small dimensions ($d \leq 6$), at the cost of additional training/calibration time. For higher-dimensional datasets, the authors acknowledge the poor performance due to curse of dimensionality requiring a very large hold-out set. Localizing the transport map seems to lower conditional coverage error.

While the empirical performance is moderate, the paper represents a meaningful application of concepts in multivariate ranking and optimal transport to multivariate conformal prediction.

**Additional Comments:**

N/A

**Audience:**

Yes

**Audience Explanation:**

This paper would be of broad interest to those working on high-dimensional predictive modeling, particularly in risk-sensitive settings where validity guarantees from conformal calibration can help.

**Claims And Evidence:**

Yes

**Claims Explanation:**

The proposed method is meaningful and interesting, and the claims scoped appropriately for the demonstrated results. The main contributions are the use of the entropic map to define the final non-conformity score from vector scores and the hard/soft localization of the map for approximate conditional coverage. The authors are careful to position these relative to concurrent work that also applies OT to multivariate conformal prediction, Thurin et al. 2025, in terms of algorithmic flexibility, computational cost, and required assumptions. The validity proofs are standard and not a part of the contributions, and the paper need not include them.

While the improvements in coverage and efficiency are only moderate in small dimensions ($d \leq 6$) and disappear for higher dimensions, the authors acknowledge these limitations and the difficulty of nonparametric map estimation in high dimensions. At the same time, I appreciated the detailed discussions of hyperparameter choices ($m$, $\epsilon$, soft vs. hard clustering) and recommendations for default settings that yield reasonable results. I believe further discussions about $\epsilon$ will help, as it's central to the choice of using the entropic map -- please see the question below.

The current presentation is very rushed and poorly formatted. Please see below for suggestions to improve clarity as well.

**Requested Changes:**

### Requested changes
- Missing related work in multivariate CP
    - Engineering a scalar score: Kuleshov et al. 2018, Messoudi et al. 2022 in addition to Johnstone and Cox 2021, Feldman et al. 2023 already cited
    - Copula-based ranking: Messoudi et al. 2021, Sun and Yu 2024 in addition to Park et al. 2024 already cited
- The presentation in Section 2.2 and the choice to follow the setup in Dheur et al. 2025 and Zhou et al. 2024 are fine, but the framing can be more general. The elementwise score $s_i$ and aggregation scheme (choice of norm to collapse the $d$ scores) are orthogonal design decisions. They can be presented more generally first, and then M-CP and Merge-CP introduced as special cases. For M-CP, $s_i$ is the CQR score and the aggregation the max. For Merge-CP, $s_i$ is the error $\hat y(x) - y$ and the aggregation the $L_2$ norm (optionally following a transformation, as in Johnstone and Cox 2021).
- The results plots are difficult to read and not all of them are referenced in the text (only Figure 2a, I believe). The color assignments to the CP algorithms also change from figure to figure, which make them difficult to interpret. It would be better if efficiency were plotted on the same page as the marginal coverage.
- Both $\varepsilon$ with $\epsilon$ are used to refer to the entropic regularization. Please check for other notational inconsistencies, such as $N/M$ vs. $n/m$
- The mixed use of \citet and \citep is impeding reading. There are also unclosed parentheses, such as in Section 2.2. Could the authors please review the text to fix all the mechanical errors?

### Optional comments / questions
- I believe the paper will benefit from a simple schematic plot illustrating the transformation of vector scores to the spherical uniform variables via the entropic map, the norm of which is (approximately) uniform in [0, 1] and enters calibration. We can assume no clustering ($K=1$) for simplicity. There can be 2D prediction sets in the original data space plotted on the side as well using the inverse entropic map, like in Figure 7 but without clustering. This will help build intuition about the geometry-aware transformation enabled by the map as well as the shapes of the prediction sets. The latter is especially helpful to readers already familiar with copula-based or rotation-based line of work in multivariate conformal prediction.
- The authors recommend a default choice of $\epsilon=0.1$, but this was surprising to me, I think of $\epsilon$ as having units of $z^2$. Th vector scores $z$ considered here were elementwise CQR scores, which has the same units as $y$ and can thus span different scales depending on the dataset. Wouldn't it be better to adapt to the scale of the cost matrix instead or preprocess $z$ somehow so that a fixed $\epsilon$ becomes a reasonable choice?
- I may be missing something -- what is meant by "debiasing" in the sentence, "As a sidenote, we do observe that debiasing the outputs of the Sinkhorn algorithm does not result in improved results" in Section 4.2?

### References

Kuleshov, Alexander, Alexander Bernstein, and Evgeny Burnaev. "Conformal prediction in manifold learning." Conformal and Probabilistic Prediction and Applications. PMLR, 2018.

Messoudi, Soundouss, Sébastien Destercke, and Sylvain Rousseau. "Ellipsoidal conformal inference for multi-target regression." Conformal and Probabilistic Prediction with Applications. PMLR, 2022.

Messoudi, Soundouss, Sébastien Destercke, and Sylvain Rousseau. "Copula-based conformal prediction for multi-target regression." Pattern Recognition 120 (2021): 108101.

Sun, Sophia Huiwen, and Rose Yu. "Copula conformal prediction for multi-step time series prediction." ICLR. 2023.

---

> ### Author Response · Authors · 2026-03-07
>
> We thank the reviewer for the positive assessment of the paper’s core idea and for the detailed suggestions on related work, framing, hyperparameters, and presentation.
>
> > Missing related work in multivariate conformal prediction
>
> We expanded the related-work discussion in Section 2.2. In particular, the revised manuscript now includes:
> - ellipsoidal / Mahalanobis approaches such as Messoudi et al. (2022),
> - manifold / Jacobian-based score engineering such as Kuleshov et al. (2018),
> - copula-based multivariate calibration approaches including Messoudi et al. (2021) and Sun and Yu (2023).
>
> We also position these approaches more explicitly relative to OTCP. In particular, copula-based approaches address dependence across multiple scalar scores, whereas OTCP addresses the complementary setting where the score is itself inherently vector-valued and must first be ordered / scalarized.
>
> > Framing of Section 2.2
>
> We rewrote the beginning of Section 2.2.1 to explicitly separate:
> -  the construction of a vector-valued discrepancy S(x,y)\in\mathbb{R}^d,
> - the scalarization rule \phi:\mathbb{R}^d\to\mathbb{R}.
> This now provides a common framework for M-CP, Merge-CP, and OTCP, which was one of the reviewer’s main suggestions.
>
> > Figures and empirical readability
>
> We improved the figure captions and made them more interpretive, and we also made method naming and discussion more consistent across the empirical section. That said, we agree this point is only partially addressed in the current revision. In particular, we have not yet added the compact aggregate empirical summary table suggested by the reviews. We agree this would improve readability further and plan to add it in the final version.
>
> > Notation consistency
>
> We performed a substantial notation cleanup. In particular, the revised manuscript now distinguishes more consistently between the population map $T^\star$, the entropic map $T_\varepsilon$, and the scalar score used in calibration. The notation in Sections 2.4 and 3.1 is now much more consistent than in the previous draft.
>
> > Citation style and mechanical issues
>
> We carried out a broad cleanup pass on citation formatting, grammar, and typography. We believe the current revision addresses the major presentation issues raised in the review.
>
> > Schematic figure for intuition
>
> We agree that a simple pipeline schematic would be very helpful. The current revision strengthens the geometric discussion and retains the visualization in Figure 7, but it does not yet include the standalone global OTCP schematic suggested in the review. We will add such a figure in the camera ready version.
>
> > Default choice of $\varepsilon$ and scale dependence
>
> We expanded the hyperparameter discussion and now explain more clearly the complementary roles of $m$ and $\varepsilon$. At the same time, we agree that a fully scale-adaptive prescription for choosing $\varepsilon$ (for example via score normalization or cost-relative tuning) is not yet part of the manuscript. We agree this would strengthen the practical guidance and we are considering it.
>
> > Clarification of the “debiasing” remark
>
> This point is now clarified in the revision. The hyperparameter section explains that the tested debiasing procedure corresponds to centering corrections for regularized transport maps, and reports that we did not observe consistent gains in the conformal metrics considered in the paper.
>
> We thank the reviewer again for the constructive comments, which led directly to several important improvements in framing and presentation.

---

### Decision · Action_Editor_Jwcb · 2026-04-01

**Recommendation:** Accept as is

**Audience:**

Yes

**Audience Explanation:**

Handling vectors of scores is a non-trivial problem that arises naturally in the context of conformal prediction (and uncertainty quantification more generally); the paper is well-written and the results are informative.

**Claims And Evidence:**

Yes

**Claims Explanation:**

This paper addresses the challenging problem of doing conformal inference when dealing with multivariate quantities, and the authors' main claims are that: (1) they propose a new technique centered around mapping vectors of scores to scalars using optimal transport insights which retains coverage guarantees, and (2) they carried out empirical tests evaluating the computational/statistical characteristics of their proposed procedure. These claims are clear, and given the content of the paper, they are solid. While there is some disagreement among the reviewers regarding novelty and significance, all the reviewers are in agreement on the point of clarity and solid claims, and the authors made revisions to more carefully compare their work with concurrent work in the literature.